# Strained two-dimensional tungsten diselenide for mechanically tunable exciton transport

Jin Myung Kim[1,2,9], Kwang-Yong Jeong[3,9], Soyeong Kwon [2,9], Jae-Pil So[4,5,9], Michael Cai Wang [6,7], Peter Snapp[6], Hong-Gyu Park [4] ✉ & SungWoo Nam [2,8] ✉

Tightly bound electron-hole pairs (excitons) hosted in atomically-thin semiconductors have emerged as prospective elements in optoelectronic devices for ultrafast and secured information transfer. The controlled exciton transport in such excitonic devices requires manipulating potential energy gradient of charge-neutral excitons, while electrical gating or nanoscale straining have shown limited efficiency of exciton transport at room temperature. Here, we report strain gradient induced exciton transport in monolayer tungsten diselenide ($WSe_2$) across microns at room temperature via steady-state pump-probe measurement. Wrinkle architecture enabled optically-resolvable local strain (2.4%) and energy gradient (49 meV/μm) to $WSe_2$. We observed strain gradient induced flux of high-energy excitons and emission of funneled, low-energy excitons at the 2.5 μm-away pump point with nearly 45% of relative emission intensity compared to that of excited excitons. Our results strongly support the strain-driven manipulation of exciton funneling in two-dimensional semiconductors at room temperature, opening up future opportunities of 2D straintronic exciton devices.

Two-dimensional (2D) van der Waals (vdW) semiconductors, including transition metal dichalcogenides (TMDs), have garnered substantial attentions during past decade owing to unique spin-valley coupling and enhanced light-matter interactions in reduced dimensionality. In particular, excitons hosted in 2D semiconductors have emerged as an essential component in ultrafast, energy-efficient optoelectronic circuits and interconnects working at ambient conditions[1,2] because of enhanced binding energy and stability even at room temperature[3,4] and rich quantum information it can carry as demonstrated in valleytronics[5,6] and antibunched photon emission[7,8]. The overarching

requirement for development of such 2D excitonic devices is to realize localization and controlled transport of exciton for information transfer. However, because of charge-neutral nature of bosonic excitons, electrical modulation of exciton flux is thus far limited to particular exciton subtypes such as charged trions[9,10] or interlayer excitons with out-of-plane dipole moments[6,11–13].

On the other hand, heterogeneous strain engineering via elastic deformation is a universally applicable technique to manipulate exciton flux because local strain directly tunes the electronic band structure, and results in built-in potential in the form of a bandgap gradient,

[1]Department of Materials Science and Engineering, University of Illinois at Urbana-Champaign, Urbana, IL, USA. [2]Department of Mechanical and Aerospace Engineering, University of California, Irvine, Irvine, CA, USA. [3]Department of Physics, Chungnam National University, Daejeon, Republic of Korea. [4]Department of Physics and Astronomy and Institute of Applied Physics, Seoul National University, Seoul, Republic of Korea. [5]Department of Physics, Soongsil University, Seoul, Republic of Korea. [6]Department of Mechanical Science and Engineering, University of Illinois at Urbana-Champaign, Urbana, IL, USA. [7]Department of Mechanical Engineering, University of South Florida, Tampa, FL, USA. [8]Department of Materials Science and Engineering, University of California, Irvine, Irvine, CA, USA. [9]These authors contributed equally: Jin Myung Kim, Kwang-Yong Jeong, Soyeong Kwon, Jae-Pil So. ✉e-mail: hgpark@snu.ac.kr; sungwoo.nam@uci.edu

which funnels excitons to regions with lowest energy without charges or aligned dipoles[14-17]. Strain-induced exciton drift/funneling was first experimentally demonstrated in irreversibly deformed bulk semiconductors at cryogenic temperatures for excitonic phase transition such as Fermi liquid or Bose-Einstein condensate (BEC)[18-20]. In contrast, exciton funneling in 2D semiconductors can be achieved at ambient temperatures because of reduced dielectric screening and resultant strong exciton binding energy[3,4]. Additionally, the unique mechanical features arising from the atomic thinness of 2D semiconductors, such as high in-plane strength and low bending stiffness, enable strain-sensitive and reversible strain-exciton coupling[21-23]. Upon these advantages and theoretical prediction in strain engineered atomic materials[14,24], strain-induced exciton funneling has been investigated in 2D semiconductors deformed by nano-indentation[25,26], three-dimensional (3D) nanostructured substrates[17,27] and trapped gas[28,29], or buckle-delaminated by substrate shrinkage[15,16,30-32]. However, experimental demonstration of 2D exciton funneling has been mostly limited to indirect evidence such as photoluminescence (PL) enhancement[15,17,29], comparison of spectral shifts[16] or spatial profiles[30], and PL emission center shift in spatiotemporally resolved PL[27]. In addition, recent studies of nano-indented TMDs also showed either exciton-trion conversion or limited exciton funneling under nanoscale local strain[25,26]. Thus, it is unclear if limited observations of strain-induced 2D exciton funneling is a result of the inherent factors, such as short ambient lifetime or trion conversion of 2D intralayer excitons, or extrinsic factors such as insufficient strain and sub-diffraction strain gradient. Accordingly, quantitative elucidation of how efficiently strain can induce exciton funneling and what physical parameters should be considered for practical design of straintronic optoelectronic devices remains elusive thus far.

Here, we present strain-engineered exciton drift in monolayer tungsten diselenide (WSe$_2$) using pump-probe measurements at ambient temperature. We employed 3D soft wrinkle architecture to create optically resolvable and mechanically reconfigurable lateral strain gradient from the apex (tensile) to the valley (compressive) of WSe$_2$ wrinkles (Fig. 1a). Straintronic switching was made possible by tailoring exciton motion from isotropic diffusion to directional drift. We observed strain-induced exciton drift over micrometer-scale distances with high transport efficiency demonstrated by pump-probe characterization. We established a theoretical strain-exciton coupling model based on time-resolved PL (TRPL) measurement and simulation, which revealed strong influence of strain gradient on exciton decay and transport dynamics. We further demonstrated a theoretical model of straintronic exciton router functional at GHz frequency. Our work presents new opportunities for exploring exciton transport and localization for straintronic devices[33] and excitonic phase transitions[6,18,19,34,35].

## Results and discussion

### Controlled heterogeneous straining of wrinkled 2D semiconductors

Our wrinkle architecture consists of an elastomeric substrate with multiple functional hetero-layers (Fig. 1a, see "Methods" for detailed fabrication). Exfoliated monolayer WSe$_2$ was transferred to a pre-stretched silica/polydimethylsiloxane (PDMS) substrate with poly-methyl methacrylate (PMMA) encapsulation, and the PDMS was slowly released to form wrinkles in WSe$_2$ due to strain-induced wrinkle surface instability[36] (Supplementary Fig. 1a). We used a stiff silica skin layer to guide conformal deformation of the WSe$_2$ during substrate contraction while exhibiting stronger vdW interaction with WSe$_2$ than clean elastomeric substrates[37,38]. We further utilized encapsulating PMMA top layer to prevent slippage or delamination of the WSe$_2$ layer which can reduce local strain.

We first investigated the geometry-dependent local strain modulation of our wrinkled WSe$_2$ structure. As shown in the height profile (Fig. 1b) and optical microscope images (Supplementary Fig. 1b), a

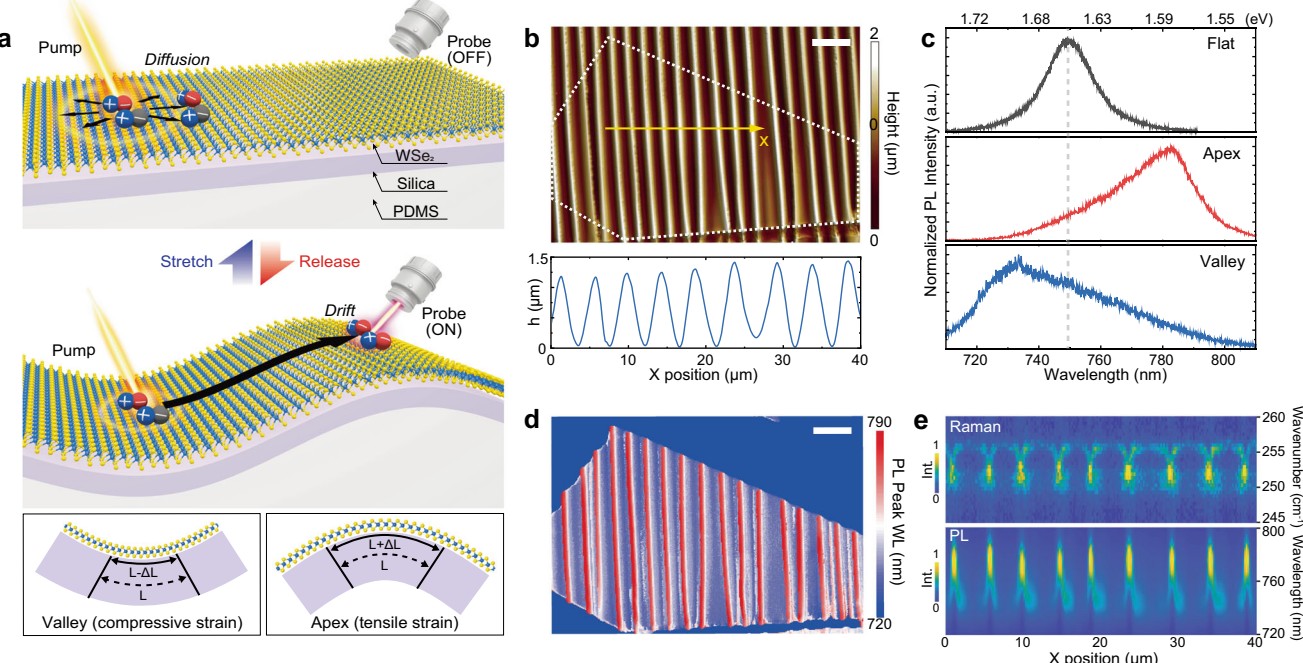

**Fig. 1 | Heterogeneous strain engineering of exciton transport in monolayer WSe$_2$ via soft wrinkle architecture. a** Schematic diagram of mechanically reconfigurable wrinkle architecture made of multiple thin films on PDMS including silica skin layer, WSe$_2$, and encapsulating PMMA layer (not shown in the schematic). Tensile and compressive strain is applied to WSe$_2$ at apex and valley of the wrinkle architecture, respectively. **b** Topography mapping of wrinkled monolayer WSe$_2$ (dotted outline) and height profile across the yellow line. **c** Normalized PL emission spectra measured from flat WSe$_2$ (top) and wrinkled WSe$_2$ at apex (middle) and valley (bottom). **d** PL peak wavelength mapping. The scale bars in (**b**, **d**) are 10 μm. **e** (Top) Raman line mapping along the yellow line in (**b**). (Bottom) PL line mapping along the same line.

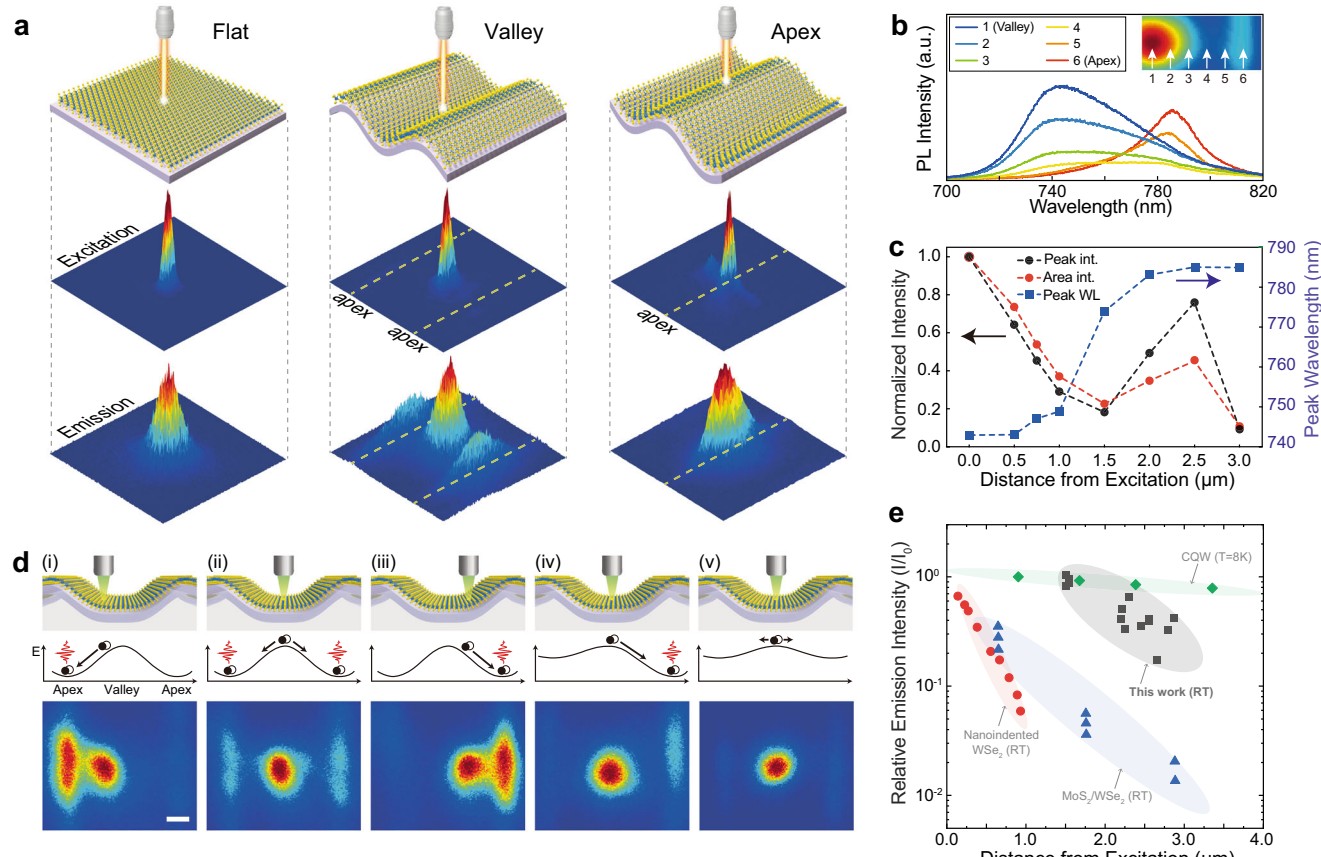

**Fig. 2 | Pump-probe characterization of exciton funneling under local strain gradient. a** Schematic of excitation points (top), measured excitation maps (second row), and measured emission maps (third row) from flat WSe₂ (left) and wrinkled WSe₂ excited at valley (center) and apex (right). The size of the excitation/emission maps is $8 \times 8\ \mu m^2$. **b** Measured PL spectra of wrinkled WSe₂ with excitation fixed at valley and emission detected from valley to adjacent apex. The distance from 1 (valley) to 6 is 2.5 μm with 6 being apex, and the spacing between consecutive numbers is 0.5 μm. **c** Emission characteristics of funneling excitons in terms of measured PL peak intensity (black), measured fluorescence (area integrated PL; red), and measured PL peak wavelength (right y-axis; blue). **d** Pump-probe measurements of wrinkled WSe₂ with different pumping position and local strain distribution. (i)–(iii) maps were measured in a same sample with varying the pumping spots. The asymmetric strain in (iv) was 0.63% (left apex) and 1.28% (right apex), while weak strain in (v) was 0.81% on both side apexes. Scale bar, 1 μm. **e** Comparison of pump-probe funneling efficiencies in various material systems. Energy gradient was created by strain (this work (black) and nanoindented WSe₂ (red)[25]) or electric field (interlayer exciton in MoS₂/WSe₂ heterostructure (blue)[12] and GaAs/AlAs coupled quantum well (green)[43]).

periodic wrinkle geometry was formed by releasing prestrain. The average height and wavelength of the wrinkles were measured to be $1.25 \pm 0.08\ \mu m$ and $4.84 \pm 0.64\ \mu m$, respectively. It is noted that overall shape of the wrinkles was sinusoidal, while radius of curvature was smaller at apex than at valley (Supplementary Fig. 1c). In correlation with the geometry, we measured PL emission of flat WSe₂ and wrinkled WSe₂ at apex and valley, and plotted normalized PL spectra in Fig. 1c. It clearly shows that PL emission was red-shifted at apex ($\Delta E = -72$ meV) and blue-shifted at valley ($\Delta E = 36$ meV) relative to the emission at flat WSe₂ ($E = 1.655$ eV). The local strain at apex and valley was calculated to be 1.47% tensile strain and 0.73% compressive strain on the basis of uniaxial strain sensitivity of monolayer WSe₂ A exciton (49 meV/% uniaxial strain)[39]. The different polarity of strain applied on WSe₂ were attributed to the opposite curvature at the apex and valley as shown in Fig. 1a. Theoretical strain on WSe₂ was estimated to be approximately ±0.69% at apex and valley based on mechanical model and experimental parameters (details in Supplementary Text 1). This value agrees well with compressive strain at the valley, while higher tensile strain at apex may arise from smaller radius of curvature. In addition to the PL shift induced by local strain, highly asymmetric PL spectra were observed in the strained area (apex and valley of wrinkled WSe₂) compared with that in unstrained area (flat WSe₂). This is attributed to energy gradient within the excitation area and funneling effect

possibly enhancing the asymmetric PL in case of local strain engineering (Supplementary Text 2). Furthermore, any signature of optically forbidden response, such as spin-forbidden dark exciton[40,41], was not observed in our wrinkled WSe₂, similar to previous works relying on out-of-plane deformation of WSe₂[15–17,25,27,42], possibly due to room-temperature optical measurement and a lack of plasmonic enhancement in optical response.

To investigate strain distribution across wrinkle structure, we performed hyperspectral PL imaging of the wrinkled WSe₂ and plotted the PL peak wavelength map (Fig. 1d). This map shows that the PL emission of wrinkled WSe₂ is gradually modulated in correlation with the wrinkle geometry in Fig. 1b. Furthermore, Fig. 1e displays the Raman and PL spectra sequentially obtained along the line in Fig. 1b. The spatial coincidence between Raman and PL shift indicates that the observed PL shift originates from the lattice strain. We also examined the effect of different focal plane of apex and valley on the sequential line/area scanning, which did not affect degree of peak shift in PL and Raman measurement (Supplementary Fig. 2). This PL shift was consistently observed across in 147 apexes and 140 valleys from 15 different samples (Supplementary Fig. 3). The average strain difference between apex and valley was calculated to be $2.4 \pm 0.3\%$, with 49 meV/μm of exciton energy gradient. The local strain value here exceeds those previously achieved by underlying substrate-induced strain engineering of

monolayer TMDs[17,21,27,39] and furthermore smooth strain gradient of monolayer TMDs across microns has not been reported yet. We attribute such a high local strain and strain gradient to conformal deformation of 2D layers with stiff silica layer and restriction of interfacial slip by encapsulation with PMMA. Consequently, the well-defined and consistent micrometer-scale strain offers optically resolvable strain gradient for more accurate optical characterization of exciton funneling behavior.

While intrinsic defects may naturally form in WSe$_2$ and could be deconvoluted in the PL peak at room temperature, these defects do not significantly affect our observations of neutral excitons funneling in room temperature via strain-induced band gap modulation. The dominance of A exciton resonances at room temperature, surpassing defect state peaks, has been confirmed through a comparison of room-temperature PL data with both cryogenic PL and room-temperature differential reflectance spectroscopy (Supplementary Text 2).

In addition to the controlled strain and strain gradient, our wrinkle architecture allows for tunability of local strain by tailoring wrinkle architecture both statically and dynamically. For static control, pre-strain, modulus, and thickness of the skin and encapsulating layer can be tuned. For instance, a different degree of PL energy shift was measured with varying skin layer thickness (Supplementary Fig. 4). In addition, mechanical reconfiguration of local strain was demonstrated by (re)stretching and releasing the PDMS substrate to vary wrinkle height and wavelength (Supplementary Fig. 5). A cyclic stretching test up to 1000 cycles showed that strain difference at the apex and valley returned to 92% of initial strain after 1000 cycles (Supplementary Fig. 6). Wrinkling-induced modulation of local band structure was also demonstrated in other types of TMDs (e.g., MoSe$_2$, WS$_2$), showing similar trends of PL shifts at apex and valley of the wrinkle structure. (Supplementary Fig. 7). These results indicate that our soft wrinkle architecture enables structural tunability and mechanical reconfigurability of neutral excitons by controlling the local strain.

## Demonstration of exciton funneling via pump-probe PL measurement

To demonstrate strain-induced exciton drift, we performed steady-state pump-probe PL characterization, as shown in bulk indirect excitons[18,19] and 2D interlayer excitons[6,12,13]. We used a pump-probe PL measurement setup that consisted of two sets of 2D galvo scanning systems allowing individual control of the pump and probe positions (Supplementary Fig. 8). While the pump laser was fixed at a specific position, reflection or fluorescence signals were collected over the (different) area of interest. Our pump-probe PL measurement allows direct visualization of exciton funneling, and no additional spectroscopic interpretation, such as PL intensity amplification or Raman/PL peak shifts, is required. The intensity profiles of the excitation laser exhibited the submicron-sized isotropic spots with a full-width at half maximum (FWHM) of ~0.5 μm (second rows in Fig. 2a). Our approach suggests negligible impact of the surface curvature in our pump-probe measurement because the period of the wrinkle structure (5 μm) is much larger than the excitation spot size (~0.5 μm) and the aspect ratio of the wrinkle height and period is relatively small (1/4 to 1/5). Furthermore, we obtained reflection images in each measurement to confirm that unintended regions of wrinkled WSe$_2$ are not pumped by reflected light. In addition, when the PL intensity was integrated and normalized, the surface curvature for different spots is not considered because the surface curvature of WSe$_2$ is slowly varying and thus, the collection efficiencies are almost same even if the excitation spots are at the flat, apex or valley WSe$_2$.

We then measured the emission profiles from the flat and wrinkled WSe$_2$ (third rows in Fig. 2a). While isotropic emission was observed on unstrained flat WSe$_2$, excitation at the apex of a wrinkle led to an elongated emission shape along the wrinkle apex. More importantly, valley pumping revealed remarkable PL emission at

adjacent apexes. These results reveal several features of exciton funneling. First, a radial diffusion occurred in the unstrained flat WSe$_2$ without directional drift. Second, excitons have lower potential energy at apex than the surrounding regions. Third, the exciton funneling was clearly observed during the valley-pumping, in which the potential energy gradient caused excitons generated at valley to drift toward apex and then recombine radiatively.

We further investigated the efficiency of exciton funneling by measuring the sequential PL spectra from valley to one of the nearest apexes while fixing the excitation at the valley (Fig. 2b). As the probe position moved toward apex from valley, gradual PL red-shift was observed. PL emission intensity (arbitrary units, a.u.) decreased when the distance from the excitation was <1.5 μm, but increased when probing near apex (from 1.5 to 2.5 μm). These measured emission characteristics were summarized in Fig. 2c. We observe several key features as follows. First, the measurement exhibited ~45% of relative fluorescence intensity and ~76% of relative PL peak intensity at 2.5 μm apart from excitation spot. Second, the relative emission intensity we measured from the intralayer excitons in strained monolayer WSe$_2$ (~45%) was more than an order of magnitude higher than that from the interlayer excitons electrically drifted in 2D vertical heterostructure (~3% of relative PL emission intensity at 2.5 μm apart from excited point)[12]. Third, the observed funneling length (d) is comparable to theoretical values reported for strained MoS$_2$ energy harvesters (d ~ 3 μm at $T$ = 270 K)[14]. If a similar estimation procedure is applied to our WSe$_2$ system, the calculation shows a maximum exciton funneling length of 6.2 μm, within which our experimental values for drift distance are present (see detailed calculation in Supplementary Text 3).

Furthermore, we demonstrate programmable exciton funneling in different funneling length and strain. Figure 2d shows top-view pump-probe emission maps with varying position of the pumping spot and strain distribution. First, we changed pumping position sequentially from one apex to the adjacent apex and measured excitation/emission PL maps (Supplementary Fig. 9). It manifests that pumping aside valley led to preferential exciton funneling toward the nearest apex and suppression of funneling to the second nearest apex in contrast to symmetric funneling in the case of pumping at the valley (Fig. 2d (i)–(iii)). In the case that the adjacent apexes have different degree of strain, the exciton distribution became also asymmetric (Fig. 2d (iv)), revealing preferential funneling toward apex under higher local strain. In contrast, neither exciton funneling from valley to apex nor exciton trapping along apex was observed from weakly strained WSe$_2$ ($\varepsilon$ = 0.8%) despite the similar wrinkle geometry (Fig. 2d (v) and Supplementary Fig. 9), implying that out-of-plane deformation itself does not significantly contribute to the PL emission or exciton funneling, but local strain and strain gradient play a more critical role in the PL shift and funneling behavior. The maximum funneling length in our measurement was about 2.9 μm with ~42% of relative fluorescence, and the maximum relative fluorescence was ~105% with 1.5 μm of funneling length.

We then plot the measured fluorescence (i.e., area-integrated PL intensity) of funneled exciton relative to fluorescence at pumping position as a function of funneling length, and compare it with other material systems (Fig. 2e). Our system exhibits more than an order of magnitude higher exciton drift efficiency compared with intralayer exciton drift induced by nanoscale strain gradient (nanoindented monolayer WSe$_2$)[25] and electrically-driven interlayer exciton drift (MoSe$_2$/WSe$_2$ heterostructure)[12]. Coupled quantum well exhibits high exciton drift efficiency due to long lifetime of spatially indirect exciton[43], while it can only operate at cryogenic temperature. Specifically, previous works on pump-probe PL measurement of strained 2D TMDs showed strain-induced exciton funneling only within the range of exciton diffusion length (<0.6 μm)[17,27], while our approach demonstrated guidance of exciton transport beyond the excitation area, by up to 2.9 μm of drift length. In addition, our controlled strain gradient

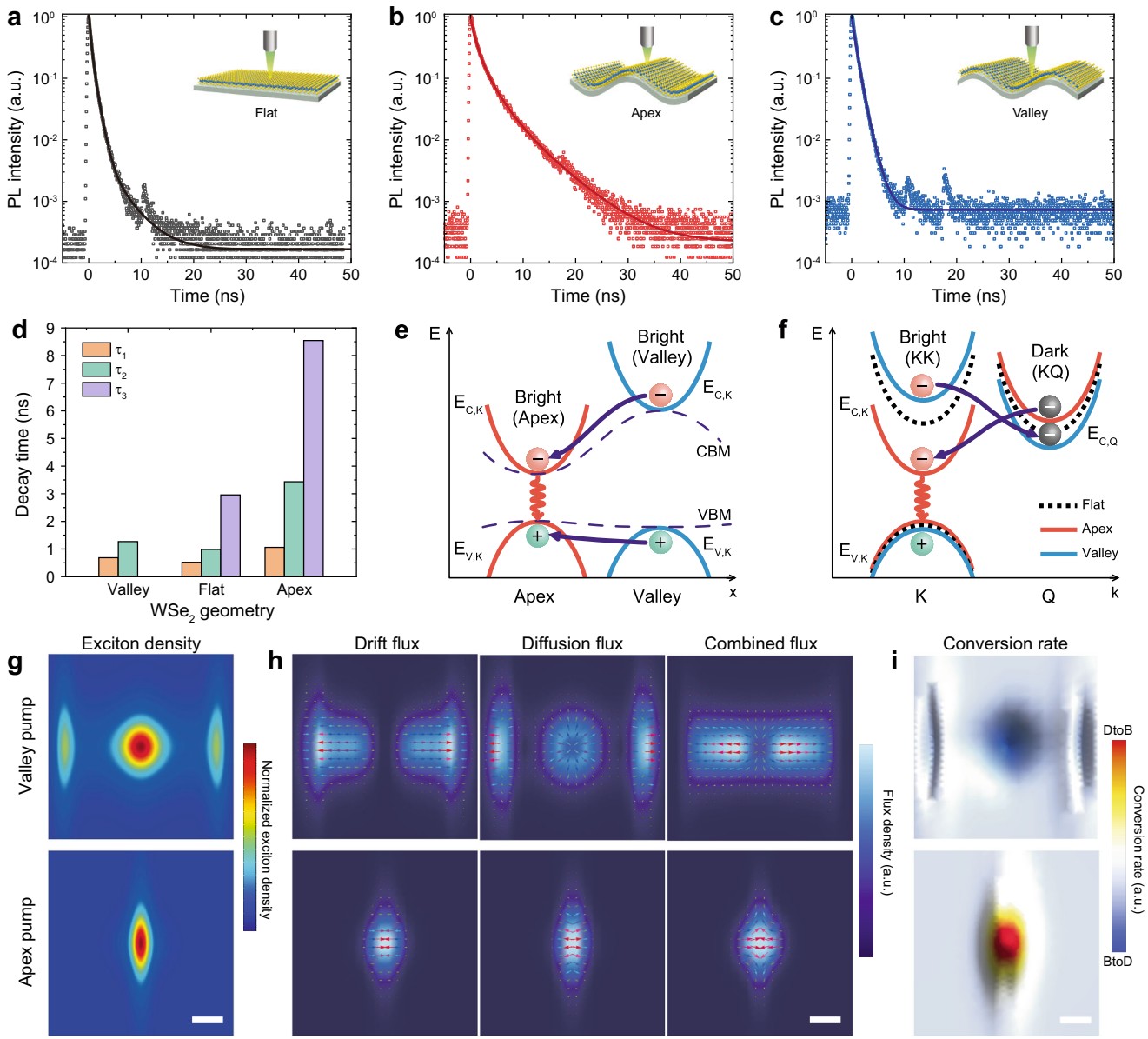

**Fig. 3 | Analyzing exciton dynamics of flat and wrinkled WSe₂ depending on the localized strain.** Time-resolved PL from flat WSe₂ (**a**) and wrinkled WSe₂ at apex (**b**) and valley (**c**). **d** Decay time constants, fitted with bi-exponential or tri-exponential equation, for flat WSe₂ and wrinkled WSe₂ at apex and valley of (**a**–**c**). **e** Schematic illustration of funneling of bright excitons from valley to apex owing to strain-induced energy gradient in real space. CBM and VBM indicate conduction band minimum and valence band maximum, respectively. **f** Schematic of generation of dark exciton at valley and its conversion to bright exciton at apex owing to different energy level of KK and KQ states under tensile and compressive strain. Simulated maps of steady-state PL emission (**g**), flux density (**h**), and bright/dark exciton conversion (**i**) for valley pumping (top) and apex pumping (bottom). Scale bars are 1 μm.

in 2D TMDs enabled the achievement of a high efficiency of exciton transport at room temperature: >40% at 2.5 μm apart from the excitation, which is more than an order of magnitude higher than that of electrically driven exciton drift[12]. Therefore, our results suggest that strain-induced exciton funneling can be an efficient and predictable approach for exciton transport at room temperature, due to the long drift length, despite relatively short lifetime of intralayer excitons. Furthermore, the funneled exciton are measured far from the excited hot lattice, implying they are thermally relaxed cold excitons, suitable for studying many-body interactions such as Fermi liquids[34] or BECs[6,18,19,34,35].

## Strain-induced modulation of exciton dynamics
Our pump-probe scanning PL measurement revealed highly efficient exciton funneling in strained WSe₂. To understand the underlying

mechanism of the exciton decay and transport dynamics under strain gradient, we performed TRPL characterizations. Figure 3a–c show TRPL spectra from the flat WSe₂ and the wrinkled WSe₂ at apex and valley regions, respectively. Compared to the exciton decay on the flat WSe₂, exciton decay at the wrinkle apex significantly slowed, while exciton decay at the valley was facilitated. We quantified the exciton decay time with exponential fitting (Fig. 3d). The longest lifetime components for flat WSe₂ and wrinkled WSe₂ at apex and valley were estimated to be 2.96, 8.54, and 1.27 ns, respectively. The lifetime in the order of nanoseconds is substantially longer than ultrafast radiative lifetime of A exciton in WSe₂ at cryogenic temperature (<5 ps)[44], which is attributed to an increase in thermalized bright exciton population and the momentum conservation that limits exciton recombination within radiative light cone. Interestingly, uniform tensile strain is known to cause faster exciton decay due to weak phonon-exciton

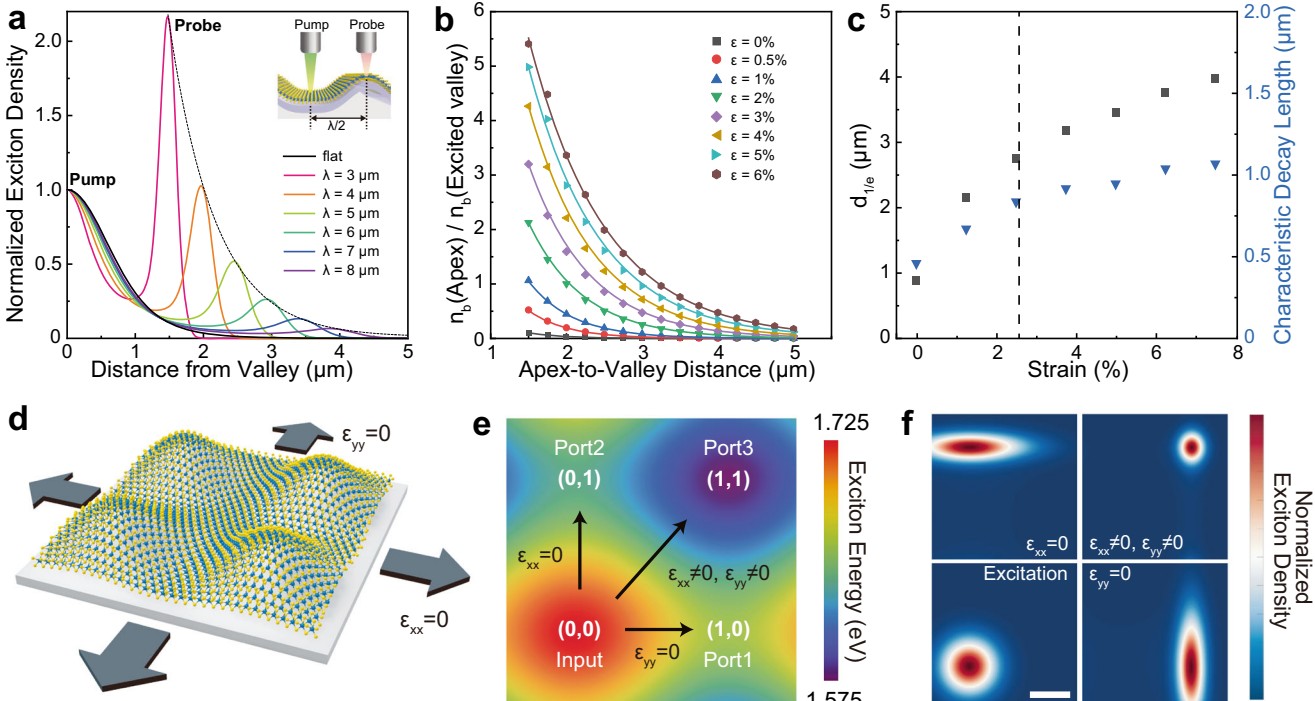

**Fig. 4 | Simulation of exciton transport and routing. a** Simulated exciton distribution normalized by exciton density at pumping position (valley). The dotted line presents exponential fitting of exciton density at half of wavelengths (apex). **b** Simulation of funneled exciton density $n_b$ at apex relative to exciton density at excited valley as functions of distance (half of wrinkle wavelength) and applied strain ε. **c** Modulation of $d_{1/e}$ (funneling distance at which density of funneled exciton at apex becomes 1/e of exciton density at excited valley) and characteristic decay length (absolute value of inverse slope of logarithmic plot of funneled exciton density at apex normalized by exciton density at excited valley). The dotted line indicates our experimental strain value (2.4%). **d** Schematic illustration of biaxially strained WSe₂ on elastomeric substrate. Stretching in x- or y- direction causes removal of local strain in the same direction. **e** Exciton energy distribution of biaxially strained WSe₂ device. The coordinate (0,0) and (1,1) indicate the valley under compressive strain and the peak under tensile strain, respectively, while (1,0) and (0,1) become local energy minima under reconfigurable stretching in y- or x-direction, respectively. **f** Simulated exciton density map of excited exciton ($t = 0$ ns) (bottom left) and drifted excitons under different local strain at $t = 1.5$ ns. Stretching in x- (y-) direction causes annihilation of x- (y-) local strain component, resulting in y-direction exciton drift (top left) or x-direction drift (bottom right), respectively. Biaxial local strain (no stretching) leads to exciton drift to (1,1) direction (top right). Scale bar is 1 µm.

coupling and destabilized Q-valley excitons in WSe₂[39]. We corroborated faster PL decay from WSe₂ under homogeneous tensile strain via two-point bending method (Supplementary Fig. 10). These results imply that localized, heterogeneous strain greatly changes exciton decay dynamics, and accurately modeling exciton decay under non-uniform strain requires taking both recombination and motion (i.e., diffusion and drift) of excitons into account in relation to strain gradient.

To explain the trend of PL decay under heterogeneous local strain, we employed a spatiotemporal exciton continuity equation:

$$\frac{\partial n}{\partial t} = -\frac{n}{\tau} - \eta n^2 + D\nabla^2 n + \mu\nabla(n(\nabla E)) \qquad (1)$$

where $n$ is exciton density, $\tau$ is radiative recombination rate, $\eta$ is exciton-exciton annihilation coefficient, $D$ is diffusivity of excitons, $\mu$ is mobility of excitons, and $E$ is potential energy of exciton, equivalent to the optical bandgap energy. To solve the continuity equation, we consider the funneling of bright excitons (i.e., electron and hole both located at K-valley in the momentum space) through bandgap energy gradient (Fig. 3e). Because of significant modulation of conduction band minimum (CBM) by applied strain, bright excitons excited at valley of wrinkle are funneled to apex where excitons can possess the lowest potential energy[14]. The greater outward flux of excitons generated at valley leads to a reduced probability of observing exciton recombination at the excitation point, which explains the faster decrease of exciton density in TRPL measurement (Fig. 3c).

Additionally, for more quantitative fitting of decay curves, we need to take strain-dependent dark excitons into account in the continuity equation. Figure 3f demonstrates how momentum-forbidden inter-valley dark exciton affects exciton dynamics under local strain. The K and Q points in the first Brillouin zone of monolayer WSe₂ are illustrated in Supplementary Fig. 11. Specifically, tensile strain shifts K-valley CBM closer to valence band (i.e., smaller bandgap) and Q-valley CBM away from valence band. For this reason, probability of phonon-assisted transition between bright state (KK) and momentum-forbidden state (KQ) varies by local strain, resulting in transition from KK to KQ excitons under compressive strain (valley) and transition from KQ to KK excitons under tensile strain (apex), thereby enhanced inflow of excitons to apex[45]. Furthermore, bright KK excitons can obtain additional drift momentum during exciton funneling, which delays or prohibits radiative decay of bright excitons[46,47]. In contrast, spin-forbidden dark excitons are expected to have negligible influence on strain-induced exciton dynamics because strain does not modulate energy difference between the spin-forbidden states and the bright states[45].

Taking the aforementioned effects into account and using physical parameters from literature and experimental conditions, we conducted numerical fitting of the continuity equation to the exciton decay curves (Supplementary Fig. 12) (see detailed calculation in Supplementary Text 4). A good agreement between the measured TRPL data and simulated fitting curves was observed in three different geometries (i.e., flat, wrinkle apex and valley), only when both bright and dark excitons were considered in the continuity equation. This

indicates that the prolonged decay time at apex is attributed to the confinement of bright exciton in the local potential trap and preferential dark-to-bright conversion due to the improved stability of bright exciton at apex.

Furthermore, we simulated the PL emission, exciton flux, and dark/bright exciton conversion using parameters obtained from the TRPL curve fitting (Fig. 3g-i). First, the simulated PL maps agreed very well with the experimental pump-probe maps, showing exciton funneling in valley-pumping and localization in apex-pumping (Fig. 3g and Supplementary Fig. 13a). In particular, the relative fluorescence intensity (i.e., drift efficiency) of valley-pumping and apex-probing quantitatively agrees with the value obtained by simulation (~41% at 2.5 μm of funneling distance) (Supplementary Fig. 13b). Second, the flux maps showing direction and density of drift, diffusion, and total flux clearly revealed a competition between drift and diffusion flux at the apex for both apex and valley pumping. It is also noted that valley-pumping showed a minimal degree of total exciton flux at the apex and strong flux along the path of exciton drift. Third, the conversion between dark and bright exciton states mainly occurs at the pumping position and the conversion direction (DtoB or BtoD in Fig. 3i) is determined by the relative energy difference between dark and bright states shown in Fig. 3f. In addition, valley-pumping shows bright-to-dark exciton conversion at the apex despite higher stability of bright exciton at the apex. This is attributed to substantial density of funneled bright exciton compared with density of dark exciton. Furthermore, there could be effects of dielectric screening induced by strained silica skin layer on the exciton transport, while its influence is expected to be less significant than the effects of strain gradient on $WSe_2$ (more discussion in Supplementary Text 5).

## Modeling of exciton-based straintronics

We carried out simulation studies of exciton funneling in terms of funneling length and strain beyond our experimental observations. The simulated exciton distribution of valley pumping and apex probing as a function of wrinkle wavelength is plotted in Fig. 4a. As the wrinkle wavelength decreases, we observe that the exciton distribution becomes narrower near the pumping position and exciton density exponentially increased at apex. The calculated relative density of funneled excitons at the apex with respect to strain also shows that higher strain gradient owing to higher applied strain or shorter funneling length significantly improves funneling efficiency (Fig. 4b). We quantified the exciton drift length using $d_{1/e}$ (funneling distance at which density of funneled exciton at apex becomes 1/e of exciton density at excited valley) and characteristic decay length (inverse slope in logarithmic plot of Fig. 4b), which shows an increase in funneling length as applying higher strain (Fig. 4c). Interestingly, our calculation demonstrated that contribution of dark-to-bright exciton conversion to total exciton funneling was less than 10%, which indicates that drift of bright exciton is a dominant mechanism of exciton funneling (Supplementary Fig. 14). Further modification of wrinkle wavelength and strain (e.g., degree of local strain, asymmetry, biaxial strain) based on the theoretical prediction of exciton funneling dynamics will provide deeper insights for design of exciton-based straintronic devices[33].

To further explore modeling of straintronic device, we propose a theoretical model of straintronic exciton router on the basis of our experimental results and analysis on strained $WSe_2$. Straintronic exciton device may require controllable exciton routing, which converts input light to excitons, programmably manipulates exciton flux to specific direction, and outputs signal as a form of photon that is created by recombined excitons[11,48]. In our model, $WSe_2$ is conformally deformed as checkerboard-shape and exerted with biaxial tensile/compressive strain at the peak/valley of the structure (Fig. 4d, e). The exciton transport directions and consequent emission port are determined by reconfigurable local strain gradient by applying external strain (detailed description of the routing model in Supplementary Text 6). The normalized exciton density at $t = 0$ ns and $t = 2$ ns is simulated and plotted in Fig. 4f. Excitation at the valley of the structure creates high energy excitons at $t = 0$ ns, which flow toward either (1,0), (0,1), or (1,1) direction based on the local strain distribution. The rate of exciton transport can be quantified by exponential time constant at the rising edge ($\tau_e$) and 10/90 rise time ($\tau_{10/90}$)[49] for which input excitation is converted to maximum output emission at the target position. It is noted that the strain-induced exciton transport can be fast enough to achieve a response of output emission photon in nanosecond timescale (i.e., GHz transmission frequency) (Supplementary Fig. 15). We attribute the rapid exciton transport to high mobility of charge-neutral excitons (300–576 $cm^2$/Vs)[27,50] compared with low diffusivity and mobility of interlayer excitons in 2D vdW heterostructures due to trapping in moiré potential[51–53] as well as smooth energy gradient along the exciton transport path.

In summary, we demonstrated efficient and controllable exciton transport in strained $WSe_2$ via soft wrinkle architecture which enables high local strain and optically resolvable strain gradient. Pump-probe and TRPL measurements manifested efficient exciton funneling with micrometer-scale funneling length and strain-dependent exciton decay. We established a straintronic device model based on the time-resolved characterization and theoretical calculation. Our results not only open up opportunities for developing mechanically reconfigurable straintronic quantum devices, but also provide a platform for heterogeneous elastic strain engineering of diverse 2D materials for novel functionalities, including tunable magnetism, anisotropic thermal conductivity, and hydrogen evolution reaction, as a function of dynamically tunable strain gradient.

## Methods
### Sample fabrication

We prepared monolayer tungsten diselenide ($WSe_2$) flake via mechanical exfoliation from single crystalline bulk $WSe_2$ (HQ Graphene, Netherlands). Specifically, we used 300 nm-thick $SiO_2$/Si wafers as exfoliation substrate. Before exfoliation, polyacrylic acid (PAA) film (about 100 nm thickness) was deposited on $SiO_2$/Si substrate as a sacrificial layer for wet transfer by spin-coating 5 wt% of PAA solution (MW ~ 50,000, Polyscience Incorporated) for 60 s at 5000 rpm. After finding 1L-$WSe_2$ flakes via optical contrast and photoluminescence (PL) spectrum, PMMA (950PMMA A2, MicroChem) was spin-coated on $WSe_2$/PAA/$SiO_2$/Si for 30 s at 6000 rpm. Separately, we prepared a PDMS substrate by mixing base elastomer and curing agent (Sylgard 184, Dow Corning Incorporation) with 10:1 weight ratio and curing at 70 °C for 60 min. Then, we prestretched a PDMS substrate with 20% of prestrain, followed by oxidizing PDMS surface with oxygen plasma treatment for 180 s at 150 mTorr pressure and 250 W of plasma power to form a stiff silica skin layer on top of PDMS. To transfer $WSe_2$ onto silica/PDMS substrate, PMMA/$WSe_2$ was separated from the wafer by floating it on the surface of DI water by dissolving the PAA layer. The separated PMMA/$WSe_2$ film was cleaned by DI water and transferred to the prestretched silica/PDMS substrate. After drying it in ambient condition for more than 48 h, we slowly removed prestrain of PDMS. Detailed fabrication process is illustrated in Supplementary Fig. 1a. For two-point bending measurement, we transferred PMMA/$WSe_2$ on 500 μm-thick polycarbonate (PC) substrate. For $MoSe_2$ and $WS_2$ wrinkles, the same fabrication process was applied with exfoliated monolayer flakes (HQ Graphene, Netherlands).

### Structural and optical characterization

Geometry of wrinkle structure was characterized by atomic force microscope (AFM) (Cypher, Asylum Research, CA, USA) and 3D laser scanning confocal microscope (VK-ZX1000, Keyence, Japan). The thickness of oxidized silica layer on PDMS was measured by X-ray reflectometry (XRR) (PANalytical Philips X'pert MRD diffractometer, USA) using Cu Kα radiation ($\lambda = 0.154056$ nm)[52]. Photoluminescence

(PL)/Raman point spectra and line profiles were obtained by Raman confocal imaging microscope (LabRAM HR, Horiba, Japan) and PL area map was measured by confocal Raman microscope (Nanophoton Raman 11, Nanophoton Corporation, Japan) equipped with line illumination. All the PL/Raman measurements were performed with 532 nm excitation laser. For reflectance spectroscopy white light from a halogen lamp was illuminated to the sample through an iris aperture and the reflected light was collected by galvo-mirror coupled confocal optics system (XperRAM, Nanobase, South Korea). Differential reflectance ($\Delta R/R$) was calculated as $(R_{WSe2+substrate} - R_{substrate})/R_{substrate}$, where $R_{WSe2+substrate}$ and $R_{substrate}$ are the reflectance of $WSe_2$ on the substrate and the bare substrate, respectively. TRPL and pump-probe measurements were carried out using a home-built fiber-coupled confocal microscope (Supplementary Fig. 8). The setup was composed of two sets of 2D galvo mirror scanning systems in order to control pump and probe positions respectively. A pulsed red laser (632 nm) was used for excitation in TRPL measurement and a continuous green laser (532 nm) was used for excitation in pump-probe measurement. The beam was focused through a 100× objective with a numerical aperture (NA) of 0.9. The detection signal was focused into a single mode fiber. A fiber splitter then led the coupled signal either to a spectrometer (Acton SpectraPro™, Princeton Instrument Inc.) for spectral measurements or to an avalanche photodiode (APD; Excelitas SPCM AQRH 13). The time-correlated-single-photon-counting (TCSPC, PicoHarp300) was used to obtain the time resolved PL. For low-temperature PL measurement, the sample was mounted in a vibration-isolated cryostat cooled by compressed helium and a 50× microscope objective lens (NA 0.42) was used for confocal PL setup similar to the room temperature measurement.

## Data availability
The main text figure data and supplementary data sets generated and analyzed in this study are available upon request to the corresponding authors.

## Code availability
The custom code developed for theoretical simulation of exciton funneling is available from the corresponding authors upon request.

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

## Acknowledgements

S.N. gratefully acknowledges support from the AFOSR (FA2386-17-1-4071, FA2386-21-1-4129 and FA9550-23-1-0134), NSF (MRSEC DMR-1720633, CMMI-2135734, CMMI-2306039, ECCS-2201054, and CBET-2035584), and ONR (N000142412533). H.-G.P. acknowledges support from the National Research Foundation of Korea (NRF) grant funded by the Korean government (MSIT) (2021R1A2C3006781 and 2021K1A3A1A32083826), the Institute of Information & Communications Technology Planning & Evaluation (IITP) grant (2020-0-00841), and the Samsung Science and Technology Foundation (SSTF-BA2401-02). K.-Y.J. acknowledges support from the National Research Foundation of Korea (NRF) grant funded by the Korean government (MSIT) (2022R1C1C1008462). M.C.W. acknowledges support from the Natural Sciences and Engineering Research Council of Canada (NSERC) PGS-D.

## Author contributions

J.M.K., K.-Y.J., S.K., and J.-P.S. contributed equally to this work. J.M.K. and M.C.W. prepared the samples, and J.M.K., K.-Y.J., S.K., and J.-P.S. performed optical characterizations. S.N. and H.-G.P. conceived and supervised the work. The manuscript was written by J.M.K., K.-Y.J., P.S., H.-G.P., and S.N. with comments and inputs from all authors.

## Competing interests

The authors declare no competing interests.
