## [Transparent Peer Review file · Nature Communications]

Strained two-dimensional tungsten diselenide for mechanically tunable exciton transport

Corresponding Author: Professor SungWoo Nam

Version 1:

Reviewer comments:

Reviewer #2

(Remarks to the Author)

I greatly appreciate the authors' efforts to address my previous comments/questions, along with those from other reviewers. The authors have gone to great lengths doing so and have addressed most of my questions.

However, I still have one major concern. The fact that the low-temperature PL of the flat region of WSe₂ in Fig. S15 is troubling. Not only the PL peak is at lower energy than the A exciton, but it is also extremely weak, and the line shape is really weird. It suggests extremely poor quality of the WSe₂ at the flat region and significant contribution of defects. This interpretation would be consistent with the very long lifetime the authors observed at room temperature (Table R1). Is it possible that all the observations reported in this work are simply the strain modulation of the defect states in WSe₂? Where would all these defects come from? Over the last few years, there have been significant developments in WSe₂ crystal growth, and the quality could be very high. Could the authors prepare one monolayer WSe₂ with PMMA capping layer but no strain and then check the PL at low temperature? How would this PL compare with previous reports, and could authors explain the low-temperature PL observed in S15?

Reviewer #3

(Remarks to the Author)

The authors have addressed most of the points raised in my referee report very thoroughly. However, I believe that some of the points have not been addressed successfully:

1) Point #1. Regarding the discussion about the use of straintronics in the title, I still believe that its use is somewhat overclaiming. In my view what they are mostly doing is strain engineering not straintronics, straintronics requires the fabrication of a device exploiting strain tunability.

2) Point #3. The authors proposed the change: "...The local strain value here exceeds those previously achieved by **local** strain engineering of monolayer TMDs^{4,14,46,71}. But the strain in [Nano Lett., 16, 5836 (2016)] is also local (see the supporting information of that manuscript, for example). So their claim about the larger reported local strain is not accurate.

Version 2:

Reviewer comments:

Reviewer #2

(Remarks to the Author)

The authors have obtained low temperature PL of unstrained WSe₂, with and without PMMA. I found that the analysis and

discussion of the data did not support the authors' claim. Here are my comments:

1. The authors should plot the low temperature PL from unstrained WSe₂ without and with PMMA on the same panel (panels c and d). With the current layout, it is hard to see the difference. But even with the current layout, it is clear that the WSe₂ with PMMA has more significant PL peaks at the lower energy. This strongly contradicts the authors' claim "These trends of temperature-dependent PL emission were not affected by presence of a PMMA encapsulating layer, indicating negligible doping effect of polymer capping."
2. The authors mentioned a couple of possibilities to explain the low temperature PL. However, it was not clear which one was which. The cited possibilities are in eV while the PL was plotted as a function of wavelength, and the way it is plotted also makes it difficult to understand. Could the authors change the x-axis to the unit of eV and try to identify the exciton peak, biexciton, localized exciton, etc.? The figure also needs to be made significantly larger to show the low temperature PL clearly.
3. Is the localized exciton state due to defects? If so, how does that affect the current interpretation? The biexciton peak can potentially be separated from the exciton through an excitation power dependence study.
4. The authors should plot the 15 K data of the unstrained WSe₂, without and with PMMA, in the same panel of the pane (f). What is the nature of the enhanced PL in panel (f)? Are they free excitons, localized excitons, or other quasiparticles?
5. The authors claim the weak PL was due to low excitation power. Do the authors have an estimate of the quantum yield? It will be a better way to compare with others' work, especially the high-quality samples. Using widely used commercially available crystals does not guarantee a good sample because the crystal is likely to differ from batch to batch. In addition, the community has found that the sample quality is highly dependent on the preparation procedure.

Version 3:

Reviewer comments:

Reviewer #2

(Remarks to the Author)

I appreciate the authors' efforts to reorganize and discuss their low-temperature data. I also want to emphasize that I do appreciate the authors' idea of using strain to manipulate the bandstructure spatially to realize exciton funneling. However, the reply did not fully address my concerns, and it actually raised serious questions about the interpretation of the data. Therefore, I would not recommend the publication of this work in Nature Communications.

My main concern is that the enhanced PL is mostly from defects, affected by the strain, instead of from exciton funneling from the local bandstructure modulation by strain.

I will list my concerns to the reply first and then summarize why these concerns raise my doubts about the data interpretation.

1. In response to my comment 1, the authors did put PL spectra on the simple figure, which is much easier for comparison. However, the exciton is not labeled in S15a in the spectrum of the strained WSe₂ with PMMA (blue). Also, excitation power dependence was not studied, and the assignment of the exciton, trion, excitons, and defects are purely based on their energies. The authors might need to be careful here as these energies can be affected by the dielectric environment and strain, varying from sample to sample. Therefore, direct comparison to references might not always be correct. Finally, the PL from Fig. S15a clearly shows that the PL is dominated by the defects, as the authors also mentioned.
2. In response to my comment 3, the authors claimed that at room temperature, PL from biexcitons and defects are not relevant and PL at room temperature is mainly from bright exciton. I would disagree with that. The thermal fluctuation at room temperature is only ~26 meV, lower than the binding energy of some excitons such as biexciton. From the energy of view, some of the excitons might still be relevant. But more importantly, it is not clear why the PL from the defects can be quickly dismissed. The PL is significantly broadened at room temperature, and eventually, the defect PL cannot be distinguished from the bright exciton in the PL spectra. I suspect that defect PL contributed significantly to the room temperature PL the author observed, which seems to agree with the long lifetime the authors measured in the main text.
3. In response to my previous comment 4, Fig. S17 actually shows significantly enhanced defect PL in the strained WSe₂. This again confirms my suspicion that the enhanced PL is from strain modulation of the defects.
4. In response to my previous comment 5, the authors agree that their sample quality is not superior and has weak low-temperature PL. I disagree with the argument that sample quality is not relevant here. If the enhanced PL is from the strain modulation of the defects (PL associated with at), it will change the interpretation of exciton funneling.

In summary, based on the quality of the current data, I suspect that the most enhanced PL is from defects instead of exciton funneling. In fact, I went through the manuscript carefully again and found that most of the data can be explained by the enhanced defect PL by strain. The measurement of PL away from the excitation spot could provide helpful information, but the data also need to be analyzed carefully, considering the diffraction limit, complicated topography of the structure, and the

low quality of the WSe₂ (low mobility and low diffusion constant). For example, what will be the diffusion constant associated with the strain modulated WSe₂ and the plain WSe₂? To confirm the exciton funneling interpretation, the authors should consider performing their experiments at low temperatures, where they can separate the exciton from defects and other contributions.

Version 4:

Reviewer comments:

Reviewer #2

(Remarks to the Author)

I have read the authors' reply thoroughly, and I would say I appreciate the authors' persistence and the extra efforts in getting additional data on the reflectance spectra as well as the new data on possible low-temperature exciton funneling. Although I still have some reservations about the interpretation of defect contributions, I believe that the current manuscript contains enough information for the community, and I would recommend the publication in Nature Communications.

I still have a few comments for the authors to consider, but this does not change my recommendation and does need to be addressed in this current version.

- 1) The defect contribution at room temperature is complicated due to the broad PL width. The argument that defects do not play a major role at room temperature might not be exactly correct. The PL width is broader than the thermal fluctuations, and the defect could have communication with the exciton state.
- 2) The reflection spectra are a strong point to argue the contribution of the exciton funneling. However, if the authors look at the spectra closely, the resonance of reflection does not exactly align with the PL. Analysis of the difference could be useful.
- 3) The description of the low-temperature exciton funneling is lacking details. I sort of guessed what the authors wanted to convey, but it was not clear directly from the description.

RESPONSE TO REVIEWERS' COMMENTS

Reviewer #1 (Remarks to the Author):

This manuscript may be publishable, but revision is needed; I would like to be invited to review any future revision. The authors present interesting results about the experimental advances on strain gradient induced exciton transport and corresponding theoretical analysis. Specifically, they present new insights on how to experimentally realize strain distributed nonuniformly and nearly periodically along one direction. They also refined the theoretical model of exciton dynamics simulation, w.r.t. that reported in the previous work [Phys. Rev. B 99, 035401 (2019)]. Though the referee thinks the work brings new insights to this field and might be publishable, the following major and minor concerns should be addressed.

Our Response:

We appreciate the referee for overall positive evaluation on our manuscript. We have addressed the major and minor concerns accordingly in the following part of the response letter.

Major concerns:

1. The authors say in the introduction that ‘... demonstrated a theoretical model of straintronic exciton router functional at GHz frequency’. This claim is ambitious but not fully supported by enough evidences. The referee cannot understand how this ‘router device’ or model really works. I can image that light is the input signal, but what is the output signal? Besides, does this device requires that the applied strain along x or y directions could be switched at GHz? Is that possible or else unimportant to the ‘router’?

Our Response:

We thank the referee for this comment. An excitonic router is a device that converts input light to excitons, programmably manipulates exciton flux to specific direction, and outputs signal as a form of photon that is created by recombined excitons^{1,2}. In our work, we experimentally showed that exciton transport can be controlled by reconfigurable strain gradient with high output-to-input signal ratio at room temperature. Based on this observation and theoretical model used in the uniaxially wrinkled WSe₂, we further proposed a theoretical model of two-dimensional exciton router, putting more emphasis on the re-routing of excitons to one of output ports. Our strain-induced exciton router consists of one input port (0,0) and three output ports ((1,0), (0,1), and (1,1)) as shown in Figure 4b-c of our original manuscript. The input and output signals are incident photon and emitted photon, respectively, identical to the electrically-driven exciton routers. Exciton routing to one of three output ports is controlled by reconfiguring local strain gradient by applying external strain (**Figure 4d-e** in our revised manuscript).

Furthermore, it is important to clarify the difference between switching frequency and transmission frequency. In our model, dynamic on/off switching is not considered in the frequency calculation because local strain gradient is kept static after reconfiguration by external strain. We rather quantified the rate of exciton transport with the rising edge (τ_e) and 10/90 rise time ($\tau_{10/90}$)³ for which input excitation is converted to maximum output emission at the target position. We observed that strain-induced exciton transport can be fast enough to achieve a response of output photon emission in nanosecond timescale (i.e., GHz transmission frequency). The GHz-frequency range signal transmission is attributed to high mobility of charge-neutral excitons (300-576 cm²/Vs)^{4,5} compared to low diffusivity and mobility of interlayer excitons in heterostructures due to trapping in moiré potential⁶⁻⁸. We also note that dynamic switching of exciton routing might be also feasible by developing fast-modulating straining devices (e.g., surface acoustic wave devices for straining 2D materials)⁹.

To respond to the reviewer's comment, we have revised our manuscript.

Page 15 in the manuscript: Straintronic exciton device may require controllable exciton routing, which converts input light to excitons, programmably manipulates exciton flux to specific direction, and outputs signal as a form of photon that is created by recombined excitons^{1,2}. ... The rate of exciton transport can be quantified by exponential time constant at the rising edge (τ_e) and 10/90 rise time ($\tau_{10/90}$)³ for which input excitation is converted to maximum output emission at the target position. It is noted that the strain-induced exciton transport can be fast enough to achieve a response of output emission photon in nanosecond timescale (i.e., GHz transmission frequency) (Supplementary Fig. 14). We attribute the rapid exciton transport to high mobility of charge-neutral excitons ($300\text{-}576\text{ cm}^2/\text{Vs}$)^{4,5} compared with low diffusivity and mobility of interlayer excitons in 2D vdW heterostructures due to trapping in moiré potential⁶⁻⁸ as well as smooth energy gradient along the exciton transport path.

Page 12 in the Supplementary Information: The input and output signals are incident photon and emitted photon, respectively, identical to the electrically-driven exciton routers. In our model, dynamic on/off switching is not considered in the frequency calculation because local strain gradient is kept static after reconfiguration by external strain.

2. The experiments and theoretical analysis for exciton transport within the one-dimensional wrinkle seem systematic and robust. However, that in the checkerboard-shape configuration is not systematic and clear. Especially, can shear strain gradient appear in the checkerboard-shape configuration? In the one-dimensional wrinkle, it is safe to rule out the effect of shear strain. But for deformations nonuniform along two directions, shear and tensile strains usually appear simultaneously due to the 'compatibility conditions for strains' [Phys. Rev. B 99, 035401 (2019)]. Since this work aims to realize 'straintronics', the effect of different types of strain should be discussed. The author can estimate the strain value on the basis of uniaxial strain sensitivity of monolayer WSe₂ A exciton (49 meV/% uniaxial strain). Then, how is the strain-gap relation determined for the checkerboard-shape configuration. Would the gap shift due to biaxial strain be simply the sum of that due to uniaxial strain? The authors should explain how the checkerboard-shape configuration could be prepared. Could the method reported in [Nature Commun. 6: 7381 (2015)] be useful? The details of strain distribution and dynamics simulation for the checkerboard-shape configuration should be provided in the supporting information.

Our Response:

We thank the referee for this valuable comment. The checkerboard shape wrinkle is superposition of perpendicular uniaxial wrinkles propagating in x and y directions, which has been experimentally demonstrated by using surface-instability driven wrinkling process, similar to our uniaxial wrinkling, in several previous reports¹⁰⁻¹³. Its geometry is similar to that of MoS₂ supported on egg-box shape 3D rigid nanostructure [Nature Commun. 6: 7381 (2015)], but checkerboard shape wrinkling using soft PDMS substrate allows for mechanically reconfigurable strain tuning and exciton routing. Since strain analysis of WSe₂ on checkerboard shape wrinkles is not experimentally examined, we assume that strain distribution is determined by the topography of the underlying substrate, which is consistent with our 1D-wrinkle measurement as well as MoS₂ on 3D rigid nanostructured substrate with similar geometry¹⁴.

As the reviewer pointed out, we acknowledge that our exciton router model did not take the shear strain into account. Admittedly, there must be contribution of shear strain component to the biaxially wrinkled

structure given the complex biaxial geometry and several interfaces between constituent layers. In the literature the reviewer referred to (Phys. Rev. B 99, 035401 (2019)), the shear strain sensitivity of MoS₂ A exciton is theoretically calculated. Unlike the linear modulation of exciton energy as a function of tensile/compressive strain, shear strain tends to cause nonlinear modulation, showing larger modulation for higher strain. In the case of small shear strain (< 2%), the shear strain sensitivity was approximately -16 meV/%strain, which is about one-third of uniaxial strain sensitivity of the same A exciton (-49.4 meV/%strain)¹⁵. To include the effect of shear strain induced modulation of exciton energy, we modified our checkerboard wrinkle model as below.

In our previous checkerboard model, we assumed that the biaxial strain is a sum of x-direction strain (ϵ_{xx}) and y-direction strain (ϵ_{yy}) which are independent of each other. In the case of independent x- and y-direction strain, the second derivatives of direct strains in their orthogonal direction become zero. Thus, we ignored the shear strain component based on the compatibility equation shown in the below equation¹⁶:

$$\frac{\partial^2 \epsilon_{xx}}{\partial y^2} + \frac{\partial^2 \epsilon_{yy}}{\partial x^2} = 2 \frac{\partial^2 \epsilon_{xy}}{\partial x \partial y}$$

In our new model, we removed the assumption of independent x- and y-direction strain components, and adopted a 2D thin plate model with sinusoidal deformation, which is described by the multiplication of sinusoidal function in x- and y-direction ($w = A \sin(2\pi x/\lambda) \sin(2\pi y/\lambda)$), where A is proportionality constant and λ is distance from a peak to nearest peak.¹⁷ The bending-induced tensile/compressive strains and shear strain applied on WSe₂ are derived by below equations in the small-deflection theory:

$$\begin{aligned} \epsilon_{xx} &= -\frac{t}{2} \frac{\partial^2 w}{\partial x^2} \\ \epsilon_{yy} &= -\frac{t}{2} \frac{\partial^2 w}{\partial y^2} \\ \epsilon_{xy} &= -\frac{t}{2} \frac{\partial^2 w}{\partial x \partial y} \end{aligned}$$

The strain distribution of the checkerboard wrinkle structure is plotted in the newly added Supplementary Fig. 17a-b. To determine the exciton energy modulation from the strain map, we need to consider the strain sensitivity of exciton energy for direct biaxial strain and shear strain. For the biaxial strain, theoretical study has demonstrated that biaxial strain sensitivity of monolayer WSe₂ A exciton energy is approximately twice higher than uniaxial strain sensitivity ($\gamma_{\text{uniaxial}} = 57.6 \text{ meV}/\%$, $\gamma_{\text{biaxial}} = 115.2 \text{ meV}/\%$)¹⁸. On the other hand, the shear strain sensitivity of A exciton of monolayer WSe₂ has not been demonstrated theoretically or experimentally. Based on theoretical prediction of MoS₂ under small shear strain¹⁹, we assumed that the biaxial strain sensitivity and shear strain sensitivity of monolayer WSe₂ is twice and one-third of the uniaxial strain sensitivity, respectively. Then, we calculated the exciton energy distribution map (Supplementary Fig. 17c). We also simulated exciton transport in the checkerboard wrinkle WSe₂ with the same physical parameters of A excitons used in the uniaxial wrinkle calculation. We did not observe significant changes in the simulation results between previous model and modified model, which may be attributed to smaller strain sensitivity of shear strain compared with tensile/compressive strains.

To reflect the changes in the exciton routing simulation, we have revised Fig. 4 (p. 29 in the manuscript) and Supplementary Fig. 14 (p. 29 in the Supplementary Information), and added detailed description of our model in Supplementary Text 6 (pp. 12-14 in the Supplementary Information) with a new figure (Supplementary Fig. 17, p. 32 in the Supplementary Information).

Figure S17. Strain distribution and exciton energy modulation in the checkerboard shape wrinkle model. **a**, Distribution of direct strain component ($\epsilon_{xx} + \epsilon_{yy}$). **b**, Distribution of shear strain component. **c**, Exciton energy modulated by direct and shear strain components.

3. Claims in the manuscript should be specific and accurate. For example, in the sentence ‘excitons transported to the nearest energy minima with high transport efficiency’, ‘high transport efficiency’ is a noninformative word, did the author define the transport efficiency exactly? What is the ‘quantum’ straintronics? In view of the referee, the exciton transport obeys classical rules as described by the diffusion and drift equation. Did the authors provide detailed picture for ‘logic operation and data transfer’? The sentence ‘Our results provide strong evidence for strain-driven manipulation of exciton funneling in two-dimensional semiconductors’, seems to say that they provide the evidence for the first time. However, [Nature Commun. 6: 7381 (2015)] has already provided strong evidences. In the sentence ‘we build a theoretical model of straintronic exciton router’, it is more accurate to say ‘propose a theoretical model’.

Our Response:

We thank the referee for this comment. We have revised the manuscript to explicitly define ‘transport efficiency’ as the relative fluorescence intensity at excitation point and at funneled point. We noted high transport efficiency because the transport efficiency in strained monolayer WSe₂ (~45%) was more than an order of magnitude higher than that of the interlayer excitons electrically drifted in 2D vertical heterostructure (~3%)²⁰. We have revised the abstract to convey clear information on this.

Straintronics is an emerging field of condensed matter physics in which mechanical strain is used to modulate band structure and resulting electric, optical, magnetic, and other properties for implementation of new information devices and technologies²¹. Exploring straintronics for quantum phenomena and many-body physics, such as Bose-Einstein condensation (BEC)²², quantum emission^{23–25}, and quantum Hall effect^{26,27}, can be examples of quantum straintronics. In this manuscript, the exciton motion is mainly described by diffusion and drift, but the model is established based on the quantum effect of excitons, such as conversion between indirect dark excitons and direct bright excitons in momentum space. In addition, the applications of straintronic exciton devices are closely related to quantum mechanical devices by

increasing density of localized cold excitons in BEC and antibunched recombination of localized excitons (single photon emitters).

Regarding ‘logic operation and data transfer’, we removed ‘logic operation’ in the revised manuscript. Both experiment and theoretical model demonstrated exciton transport (i.e., data transfer) by means of reconfigurable strain engineering. The detailed description of logic operation may require more sophisticated design of straintronic devices, which we note is beyond the scope of this manuscript.

While the referred paper [*Nature Commun.* 6: 7381 (2015)] demonstrated exciton funneling in MoS₂ based on PL enhancement, there has been ongoing controversy on whether short-lived exciton can be funneled at room temperature. Recent report also showed trion formation under local tensile strain in monolayer WS₂²⁸. Thus, it was difficult to completely exclude a possibility that other exciton components might play a role in the PL intensity enhancement. In contrast, our finding of exciton funneling is based on pump-probe PL measurement, where PL emission of funneled exciton is directly measured by two sets of 2D galvo scanning systems, which can clearly support exciton funneling at room temperature with no need for additional spectroscopic interpretation. This is why we stated that our results provide strong evidence for strain-driven manipulation of exciton funneling in two-dimensional semiconductors. As the reviewer pointed out, however, this may be mistakenly viewed as we are claiming the first evidence provided. We thus have revised the sentence in the abstract to “Our results strongly support the strain-driven manipulation of exciton funneling in two-dimensional semiconductors at room temperature”, and also revised the manuscript accordingly.

Regarding the sentence ‘we build a theoretical model of straintronic exciton router’, we agree with the reviewer, and have changed the phrase to “propose a theoretical model”.

Page 2 in the manuscript: We observed strain gradient induced flux of high-energy excitons ~~transported to the nearest energy minima with~~ and emission of funneled, low-energy excitons at the 2.5 μm-away pump point with nearly 45% relative emission intensity compared to that of excited excitons. Our results ~~provide strong evidence for~~ strongly support the strain-driven manipulation of exciton funneling in two-dimensional semiconductors at room temperature, opening up future opportunities ~~for quantum straintronics and excitonic phase transitions~~ of 2D straintronic exciton devices.

Page 8 in the manuscript: Our pump-probe PL measurement allows direct visualization of exciton funneling, and no additional spectroscopic interpretation, such as PL intensity amplification or Raman/PL peak shifts, is required.

Page 15 in the manuscript: we ~~build~~ propose a theoretical model of straintronic exciton router on the basis of our experimental results and analysis on strained WSe₂.

Page 10 in the Supplementary Information: It should be noted that our calculation mainly elucidates exciton motion as functions of diffusion and drift, but the model is established based on the quantum effect of excitons, such as conversion between indirect dark excitons and direct bright excitons in momentum space. In addition, strain-induced exciton transport and localization is closely related to exploring quantum mechanical phenomena and functionalities by increasing density of localized cold excitons in Bose-Einstein condensation and antibunched recombination of localized excitons (single photon emitters). Thus, our model may pave a pathway for rational design of quantum straintronic devices based on strain engineering of atomically-thin semiconductors.

Minor concerns:

1. There are some typos and mistakes in the manuscript and supporting information. For examples, in the sentence ‘drift to (0,1) and (1,0) direction...’ on page 13, ‘direction’ should be ‘directions’. In Figs S12b and S12c, should the name of the variable be the strain difference between apex and valley? The unit for strain gradient is definitely not ‘%’. In the abstract, ‘strain-induced exciton transport’ is not accurate. It is the strain gradient that induce exciton transport.

Our Response:

We thank the referee for the detailed comments. We have revised them accordingly.

Page 12 in the Supplementary Information: By changing local strain (e.g., stretching) in x - and y -directions we can induce exciton drift to (0,1) and (1,0) ~~direction~~ directions, respectively.

Page 29 in the Supplementary Information: (name of the variable on x -axis in Fig. S14b-c) strain ~~gradient between (1,0) and (0,0)~~ difference between peak and valley (%)

*Page 2 in the manuscript: Here, we report strain **gradient** induced exciton transport in monolayer tungsten diselenide (WSe_2) across microns at room temperature via steady-state pump-probe measurement.*

2. The K and Q points in the momentum space should be illustrated, to do good to non-specialist readers.

Our Response:

We thank the referee for this comment. We have included a new schematic of K/Q point in momentum space (first BZ) in SI (**Figure S10**) and revised the manuscript.

Page 12 in the manuscript: The K and Q points in the first Brillouin zone of monolayer WSe_2 are illustrated in Supplementary Fig. 10.

Figure S10. Schematic illustration of momentum space of wrinkled WSe_2 . **a**, First Brillouin zone points of unstrained WSe_2 in momentum space. **b**, Band diagram at the apex of wrinkled WSe_2 , with lower energy of direct KK excitons. **c**, Band diagram at the valley of wrinkled WSe_2 , showing lower stability of bright exciton compared to momentum-indirect dark exciton.

3.The referee thinks that the academic writing (especially that of the abstract and introduction) should be improved.

Our Response:

We thank the referee for this comment. We have revised the abstract and introduction according to the comments.

Page 2 in the manuscript (abstract): ~~Two-dimensional (2D) van der Waals semiconductors~~ **Tightly bound electron-hole pairs (excitons) hosted in atomically-thin semiconductors** have emerged as ~~a prospective platform for ultrafast optoelectronic information processing at room temperature owing to strongly bound excitons.~~ **prospective elements in optoelectronic devices for ultrafast and secured information transfer. In 2D excitonic devices, it is critical to create lateral gradient of exciton energy** ~~The controlled exciton transport in such excitonic devices requires manipulating potential energy gradient of charge-neutral excitons,~~ while electrical gating or nanoscale straining have shown limited efficiency of exciton transport at room temperature. Here, we report strain **gradient** induced exciton transport in monolayer tungsten diselenide (WSe₂) across microns at room temperature via steady-state pump-probe measurement. Wrinkle architecture enabled optically-resolvable local strain (2.4%) and energy gradient (49 meV/ μm) to WSe₂. We observed strain gradient induced flux of high-energy excitons ~~transported to the nearest energy minima with high transport efficiency with~~ **and emission of funneled, low-energy excitons at the 2.5 μm -away pump point with nearly 45% of relative emission intensity compared to that of excited excitons.** ~~Our time-resolved characterization combined with theoretical calculation suggests potential straintronic exciton devices functional at GHz frequency. Our results provide strong evidence for~~ **strongly support the** strain-driven manipulation of exciton funneling in two-dimensional semiconductors **at room temperature**, opening up future opportunities for ~~quantum straintronics and excitonic phase transitions~~ **of 2D straintronic exciton devices.**

Page 3 in the manuscript (introduction): ~~Optoelectronic information processing is an appealing platform for ultrafast, energy efficient signal computing and optical communication beyond electronic devices^{29,30}. Excitons in atomically thin~~ **Two-dimensional (2D) van der Waals (vdW) semiconductors,** including transition metal dichalcogenides (TMDs), have ~~potential for integrated optoelectronic circuits and interconnects because they are strongly bound, stable even at room temperature^{31,32} and can carry rich quantum information as demonstrated in valleytronics^{33,34} and antibunched photon emission^{35,36}.~~ **garnered substantial attentions during past decade owing to unique spin-valley coupling and enhanced light-matter interactions in reduced dimensionality. In particular, excitons hosted in 2D semiconductors have emerged as an essential component in ultrafast, energy-efficient optoelectronic circuits and interconnects working at ambient conditions^{29,30} because of enhanced binding energy and stability even at room temperature^{31,32} and rich quantum information it can carry as demonstrated in valleytronics^{33,34} and antibunched photon emission^{35,36}.** The overarching requirement for development of such 2D excitonic devices is to realize localization and controlled transport of exciton for ~~logic operation and data~~ **information** transfer. However, because ~~of bosonic excitons are charge-neutral nature of bosonic excitons,~~ ... Strain-induced exciton drift/funneling was first experimentally demonstrated in irreversibly deformed bulk semiconductors at cryogenic temperatures **for excitonic phase transition such as Fermi liquid or Bose-Einstein condensate (BEC)^{22,37,38}.**

4.Could the dielectric screening induced by the silicon layer influence the exciton transport?

Our Response:

We thank the referee for this comment. As the reviewer pointed out, dielectric environment plays a critical role in light-matter interactions in 2D materials. Our system consists of WSe₂ supported on a silicon oxide layer, which induces strong dielectric screening effects and thus leads to weaker exciton binding energy and reduced quasiparticle bandgap compared with those of freestanding WSe₂. Here, we consider two dielectric screening factors from amorphous silica skin layer. First, dielectric screening of underlying silica layer is determined by high frequency dielectric constant. Since the high frequency dielectric constant of silicon oxide is reported to be 2.1^{39,40}, it may induce less stronger dielectric screening effect than hBN (4.5)³⁹ or sapphire (4.1)⁴¹. However, it should be noted that dielectric screening does not significantly alter the energy of 1s state exciton (i.e., A exciton) in WSe₂ at room temperature, while the optical bandgap of higher excitonic states (2s to 4s) decreases at higher dielectric screening environment⁴¹. Because the exciton transport described in this manuscript focuses on energy gradient effect on A exciton, the dielectric screening of silica layer may have limited contribution to exciton transport in terms of substrate material selection.

The other aspect we might need to consider is the gradient factor arising from local strain gradient. The local strain exerted to silica can change local dielectric constant and cause variation of dielectric screening effect. However, strain-induced tuning of the dielectric constant of amorphous silica layer requires significant degree of stress (i.e., 450 MPa of biaxial stress for 1% of variation in dielectric constant)⁴². Considering maximum local tensile strain at the apex of silica layer (<3%) and the Young's modulus of silica (70 GPa), the local variation in dielectric constant is estimated to be less than 3%. Thus, the influence of local variation in dielectric constant of silica layer is expected to be small in exciton transport, in comparison to energy modulation induced by local strain on WSe₂.

We have revised our manuscript and added discussions in the Supplementary Information (*pp. 10-12 in the Supplementary Information (Supplementary Text 5): Dielectric screening and doping effects from the encapsulating layer and the skin layer*).

Page 14 in the manuscript: Furthermore, there could be effects of dielectric screening induced by strained silica skin layer on the exciton transport, while its influence is expected to be less significant than the effects of strain gradient on WSe₂ (more discussion in Supplementary Text 5).

Reviewer #2 (Remarks to the Author):

Kim et al. presented a study on exciton funneling effects in strained WSe₂. Wrinkled structures of TMDs and the exciton funneling effects have been reported before. Here, the authors try to study quantitatively how efficiently the strain can induce the exciton funneling and what parameters are important. Although the independent control of the excitation and collection through two sets of Galvano mirrors is interesting, I found that this work lacks the novelty and significance to be published in Nature Electronics. I would suggest the submission to a more specialized journal. I also have a few specific comments that the authors probably want to address before the resubmission.

Our Response:

We thank the referee 2 for his/her review of our work. The most significant insight of our work is to achieve room-temperature exciton transport via strain engineering of 2D semiconductors. Indeed, the following features in our wrinkled TMD structure are distinguishable from previous observations: (1) 5 times longer period/wavelength of wrinkle, (2) 2.3 times higher strain, and (3) robust reconfigurability up to 1,000 cycles compared to reported TMD wrinkles [*Nano Lett.* 21, 43 (2021)]. These advances are the results of well-designed multilayer wrinkling that has not been demonstrated in previous studies. These advances are also responsible for the long-range and robust strain gradient, which has not been demonstrated before.

Furthermore, novelty and significance can be found in our optical measurement. More specifically:

- (a) Previous pump-probe PL measurements showed strain-induced exciton funneling of 2D TMDs only within the range of exciton diffusion length ($<0.6 \mu\text{m}$) [*Nano Lett.* 20, 6791 (2020)]. However, our approach demonstrated guidance of exciton transport beyond the excitation area, by up to $2.9 \mu\text{m}$ of drift length. Such a significant advance can offer a new concept of strain-induced exciton transport.
- (b) Our controlled strain gradient in 2D TMDs enabled the achievement of a high efficiency of exciton transport at room temperature: $>40\%$ at $2.5 \mu\text{m}$ apart from the excitation. Moreover, our device efficiency is more than an order of magnitude higher than that of electrically driven exciton drift [*Nature* 560, 340 (2018)].

Thus, we respectfully disagree with the reviewer that our work lacks the novelty and significance. We are happy to address the reviewer's critical remarks, important questions, and specific suggestions.

We have also revised our manuscript to underline our novelty and significance.

Pages 10-11 in the manuscript: Our system exhibits more than an order of magnitude higher exciton drift efficiency compared with intralayer exciton drift induced by nanoscale strain gradient (nanoindented monolayer WSe₂)⁴³ and electrically-driven interlayer exciton drift (MoSe₂/WSe₂ heterostructure)²⁰. Coupled quantum well exhibits high exciton drift efficiency due to long lifetime of spatially indirect exciton⁴³, while it can only operate at cryogenic temperature. Specifically, previous works on pump-probe PL measurement of strained 2D TMDs showed strain-induced exciton funneling only within the range of exciton diffusion length ($<0.6 \mu\text{m}$)^{17,27}, while our approach demonstrated guidance of exciton transport beyond the excitation area, by up to $2.9 \mu\text{m}$ of drift length. In addition, our controlled strain gradient in 2D TMDs enabled the achievement of a high efficiency of exciton transport at room temperature: $>40\%$ at $2.5 \mu\text{m}$ apart from the excitation, which is more than an order of magnitude higher than that of electrically driven exciton drift¹². Therefore, our results suggest that

strain-induced exciton funneling can be an efficient and predictable approach for exciton transport ~~in~~at room temperature, due to the long drift length, despite relatively short lifetime of intralayer excitons.

1) The studied WSe₂ is in contact with polymer and does not have BN encapsulation. What is doping of WSe₂? Is the PL from the intralayer exciton? How does it affect the exciton funneling?

Our Response:

We thank the referee for this comment. As the reviewer pointed out, we used PMMA (thickness of ~70 nm) as an encapsulating material, to avoid the undesirable effects of h-BN that may appear in our wrinkle system. For example, the use of thick and stiff h-BN (typical thickness of >30 nm) on both sides or top side of monolayer WSe₂ can cause further disturbed wrinkling because the stiffness of h-BN (~860 GPa for monolayer and few-layer hBN⁴⁴) is much higher than that of soft polymer (~3 GPa for thin film PMMA⁴⁵). In fact, when h-BN was used, we often found that no wrinkles were formed around h-BN or bubbles were trapped at the interface between h-BN and WSe₂.

Furthermore, we examined various substrates, including SiO₂, PMMA and h-BN, and measured PL emission of WSe₂ (**Figure S16**). The measurement showed that the PL peak emission wavelengths were almost same at ~750 nm (1.65 eV), which corresponds to the intralayer emission of WSe₂⁴⁶. In addition, considering similar peak wavelengths and emission shapes, we can conclude that there is no substantial doping effect or exciton funneling effect from the PMMA substrate compared with SiO₂ or h-BN.

To respond to the reviewer's comment, we have revised supplementary information (Supplementary Text 5).

Pages 11-12 in the Supplementary Information: In addition to the dielectric screening effect, there could be doping effect due to contact with a polymer layer. In this work, we used PMMA (thickness of ~70 nm) as an encapsulating material, to avoid the undesirable effects of h-BN that may appear in our wrinkle system. For example, the use of thick and stiff h-BN (typical thickness of >30 nm) on both sides or top side of monolayer WSe₂ can cause further disturbed wrinkling because the stiffness of h-BN (~860 GPa for monolayer and few-layer h-BN⁴⁴) is much higher than that of soft polymer (~3 GPa for thin film PMMA⁴⁵). In fact, when h-BN was used, we often found that no wrinkles were formed around h-BN or bubbles were trapped at the interface between h-BN and WSe₂. Furthermore, we examined various substrates, including SiO₂, PMMA and h-BN, and measured PL emission of WSe₂ (Supplementary Fig. 16). The measurement showed that the PL peak emission wavelengths were almost same at ~750 nm (1.65 eV), which corresponds to the intralayer emission of WSe₂⁴⁶. In addition, considering similar peak wavelengths and emission shapes, we can conclude that there is no substantial doping effect or exciton funneling effect from the PMMA substrate compared with SiO₂ or h-BN.

Figure S16. PL emission spectra of WSe₂ on various substrates. Flat WSe₂ is on bare 300 nm SiO₂/Si wafer, 70 nm-thick PMMA spin-coated on SiO₂/Si wafer, and h-BN exfoliated on SiO₂/Si wafer.

2) The PL from the strained area (Apex and Valley in Fig. 1c) are highly asymmetric. Could that due to different excitonic components in the PL? A low-temperature measurement will clarify that.

Our Response:

We thank the referee for this comment. As the reviewer pointed out, highly asymmetric PL was measured under local tensile (apex) and local compressive (valley) strains (Fig. 1c). Indeed, the asymmetric PL emission spectrum is usually observed even in unstrained monolayer WSe₂ due to phonon scattering of momentum indirect excitons such as KQ exciton (i.e., a hole in the K point of the valence band and an electron in the Q point of the conduction band)^{46,47}. On the other hand, uniform tensile strain causes more symmetric PL emission of WSe₂ as shown in the PL spectra of uniformly stretched WSe₂ (Supplementary Fig. 9c). This result agrees well with literature showing a decrease in PL linewidth of uniformly strained WSe₂ due to reduced stability of KQ exciton and less phonon scattering between KK and KQ excitons^{46,47}. Thus, asymmetric PL emission of locally strained WSe₂ is not likely to suggest strain-induced formation of different excitonic components (e.g., trion, biexciton). We rather attribute the asymmetry in PL spectra of wrinkled WSe₂ to (1) the energy gradient within the excitation area and (2) funneling effect which could further enhance the asymmetric PL in case of local strain engineering.

In addition, as the reviewer suggested, we performed low-temperature PL measurement (T = 15 K) of the flat and wrinkled WSe₂ prepared using the same fabrication condition (**Figure S15**). The unstrained flat WSe₂ showed relatively weak PL emission at 740 nm (1.675 eV), which corresponds to PL emission of localized/bound excitons rather than charged excitons (1.70 eV) or neutral excitons (1.72 eV)⁴⁸. In contrast, we observed sharp and strong PL emissions along apex lines of the wrinkled WSe₂ at low temperature. Particularly, the emission peaks measured at low temperature tend to be more symmetric compared with asymmetric PL spectra obtained at room temperature. These observed PL emission peaks in wrinkled WSe₂ may be attributed to excitons confined in locally strained area⁵⁰ with reduced phonon scattering at low temperature. It should be noted that the PL emission of wrinkled WSe₂ measured at low temperature does not match with emission of trion. These results suggest that the asymmetric PL emission of locally strained WSe₂ at room temperature may not result from other excitonic components, but from strain gradient induced effects such as energy gradient within the excitation area and/or exciton funneling.

To respond to the reviewer's comment, we have revised our manuscript and added new section in the Supplementary Information (page 6. *Supplementary Text 2: Asymmetric PL spectra and excitonic components*).

Page 6 in the manuscript: In addition to the PL shift induced by local strain, highly asymmetric PL spectra were observed in the strained area (apex and valley of wrinkled WSe₂) compared with that in unstrained area (flat WSe₂). This is attributed to energy gradient within the excitation area and funneling effect possibly enhancing the asymmetric PL in case of local strain engineering (Supplementary Text 2).

Figure S15. Localized PL emission at low temperature ($T = 15$ K). **a**, Fluorescence intensity map of wrinkled WSe₂, showing localized strong PL emission along the apex line. The scale bar is 5 μm . **b**, PL spectra of flat WSe₂ and wrinkled WSe₂ at localized emission points in (a).

3) For the so-called “pump-probe” measurement (separate excitation and collection measurement), how do authors consider the curvature of the surface? It can potentially reflect the normal incident light (to a flat surface) to different spots. How would authors rule out that effects?

Our Response:

We thank the referee for this comment. We expect that the effect of the surface curvature is negligible in our pump-probe measurement because the period of the wrinkle structure (5 μm) is much larger than the excitation spot size (~ 0.5 μm) and the aspect ratio of the wrinkle height and period is relatively small (1/4 to 1/5). Furthermore, we took reflection images without longpass filter in each measurement, to confirm that unintended regions of wrinkled WSe₂ are not pumped by reflected laser. We have addressed this point in the revised manuscript.

Page 8 in the manuscript: Our approach suggests negligible impact of the surface curvature in our pump-probe measurement because the period of the wrinkle structure (5 μm) is much larger than the excitation spot size (~ 0.5 μm) and the aspect ratio of the wrinkle height and period is relatively small (1/4 to 1/5). Furthermore, we obtained reflection images in each measurement to confirm that unintended regions of wrinkled WSe₂ are not pumped by reflected light.

4) Have the authors considered the surface curvature for different spots when they integrate and normalize the PL intensity? For the flat surface, the radiation from the intralayer exciton is out of the plane, while that from the spin-forbidden dark exciton is in-plane. For a curved surface, both might be collected, with different efficiency compared to the flat surface. How does this affect the measurements?

Our Response:

We thank the reviewer for this comment. We did not consider the surface curvature for different spots when the PL intensity was integrated and normalized. As we addressed in our response to Comment #3, the surface curvature of WSe₂ is slowly varying and thus, the collection efficiencies are almost same even if the excitation spots are at the flat, apex or valley WSe₂.

Optical responses of 2D TMD are highly anisotropic between out-of-plane and in-plane responses, such as dielectric tensor and surface susceptibility⁵¹⁻⁵³. As the referee pointed out, the in-plane emission of spin-forbidden dark exciton of 2D TMDs has been reported where light propagating parallel to the TMD in-plane was detected at cryogenic temperature ($T = 13$ K)⁵⁴ or when a metallic tip was placed on the TMD surface for strong plasmonic coupling at room temperature⁵⁵. Despite the possibility of detecting spin-forbidden dark exciton in a curved surface, a number of past works relying on out-of-plane deformation of WSe₂ and TMDs did not reveal any signature of spin dark exciton at room temperature^{4,14,43,56-58} due to optically forbidden out-of-plane response.

Moreover, our study revealed dominant impacts of strain gradient on exciton transport compared to out-of-plane deformation. **Figure R1** shows pump-probe emission maps and PL emission spectra of wrinkled WSe₂ with low degree of local strain at apex and valley while having similar out-of-plane deformations (i.e., wrinkle wavelength and height). The energy difference between WSe₂ at apex and valley was about 40 meV, which is smaller than those of the optimized samples (~120 meV). Because of the weak strain gradient in this sample, we observed more isotropic PL emission and less degree of exciton funneling in the valley-pumping map. This implies that the out-of-plane deformation itself does not significantly contribute to the PL emission or exciton funneling, but local strain and strain gradient play a more critical role in the PL shift and funneling behavior. Taken together, we believe in-plane emission of spin-forbidden exciton and out-of-plane deformation are not the dominant factors of exciton funneling in our study.

We have added additional discussions regarding this in our revised manuscript.

Page 6 in the manuscript: Furthermore, any signature of optically forbidden response, such as spin-forbidden dark exciton^{40,41}, was not observed in our wrinkled WSe₂, similar to previous works relying on out-of-plane deformation of WSe₂^{15-17,25,27,42}, possibly due to room-temperature optical measurement and a lack of plasmonic enhancement in optical response.

Page 8 in the manuscript: In addition, when the PL intensity was integrated and normalized, the surface curvature for different spots is not considered because the surface curvature of WSe₂ is slowly varying and thus, the collection efficiencies are almost same even if the excitation spots are at the flat, apex or valley WSe₂.

Page 10 in the manuscript: ... implying that out-of-plane deformation itself does not significantly contribute to the PL emission or exciton funneling, but local strain and strain gradient play a more critical role in the PL shift and funneling behavior.

Figure R1. Pump-probe maps and PL emission spectra of less-strained, wrinkled WSe₂. **a-c**, Pump-probe emission maps with pumping at (a) left apex, (b) right apex, and (c) valley of the wrinkle. The red dotted lines indicate left/right apex. Scale bars, 1 μm . **d**, PL emission spectra of the less-strained WSe₂ obtained at left/right apex and valley. This sample has similar structural parameters as that shown in Supplementary Fig. 8b.

5) The measured TRPI lifetime in Fig. 3 is significantly longer than the intralayer exciton lifetime at low temperatures. Could the authors explain why the intralayer exciton lifetime is increased at room temperature? How to piece out contributions from the dark excitons? Again, low-temperature measurements and temperature dependence might help.

Our Response:

We thank the reviewer for this comment. The neutral A exciton in monolayer WSe₂ shows the ultrafast radiative lifetime (< 5 ps) at cryogenic temperature, and also decays faster than other excitonic components do (e.g., trion lifetime > 18 ps at 4 K)⁵⁹. As temperature increases to room temperature, the intralayer exciton lifetime becomes longer to the order of nanoseconds due to an increase in thermalized bright exciton population and the momentum conservation that limits exciton recombination within radiative light cone. This behavior can be theoretically described using the following equation⁶⁰:

$$\langle \tau_s \rangle = \tau_s(0) \frac{3}{4} \left(\frac{E_s(0)^2}{2M_s c^2} \right)^{-1} k_B T$$

where $\langle \tau_s \rangle$ is averaged radiative lifetime, $\tau_s(0)$ is radiative lifetime for wavevector $Q = 0$, $E_s(0)$ is exciton energy for $Q = 0$, M_s is effective mass of exciton, c is the speed of light, k_B is the Boltzmann constant, and T is temperature. This equation implied the linear increase of exciton lifetime with respect to increasing temperature. Additionally, this theoretical prediction agrees well with the experimental exciton lifetimes measured by time-resolved PL (Table R1). As the reviewer pointed out, there could be contributions from optically forbidden dark excitons (e.g., spin-forbidden or momentum-forbidden transitions), while the

exciton population in the dark states is limited at room temperature due to thermalization⁶¹. For this reason, to the best of our knowledge, lifetime of dark excitons in monolayer WSe₂ has not been measured at room temperature. Further studies on the lifetime of dark excitons would help understand the exciton dynamics and exciton transport behavior of WSe₂, but this will be beyond the scope of our study. We have revised our manuscript to discuss this aspect.

Page 11 in the manuscript: The lifetime in the order of nanoseconds is substantially longer than ultrafast radiative lifetime of A exciton in WSe₂ at cryogenic temperature (< 5 ps)⁵⁹, which is attributed to an increase in thermalized bright exciton population and the momentum conservation that limits exciton recombination within radiative light cone.

Table R1. Radiative lifetime of monolayer WSe₂ at various temperature reported in literature and measured in this work.

Temperature	Radiative lifetime	Reference
4 K	4 ps	Wang et al ⁵⁹
300 K	4 ns, 12 ns (biexp.)	Mouri et al ⁶²
298 K	5.3 ns (unstrained) 3.89 ns (0.63% tensile strain) 2.54 ns (1.28% tensile strain)	Niehues et al ⁴⁶
130 K	70 ps	Yan et al ⁶³
260 K	0.25 ns	
290 K	1 ns	Zhang et al ⁶¹
298 K	0.73 ns	Leon et al ⁴
298 K	1.02 ns	Moon et al ⁴³
298 K	1.27 ns (local compressive strain) 2.96 ns (unstrained) 8.54 ns (local tensile strain)	This work

Reviewer #3 (Remarks to the Author):

The manuscript "Strained two-dimensional semiconductors toward straintronics" by Hong-Gyu Park, SungWoo Nam and co-workers present a very interesting study of rippled single-layer WSe₂ with scanning confocal PL and Raman. The authors transferred 1L WSe₂ onto pre-stretched PDMS in order to achieve a sinusoidal ripple pattern that leads to alternating compressive/tensile strains on the flake. This method to achieve this rippled structure is well-known in the thin-film community (e.g. <https://doi.org/10.1038/nmat1175>) and it has been recently used to strain engineer 2D materials (e.g. <https://doi.org/10.1021/acs.nanolett.5b04670>, <https://doi.org/10.1088/1361-6528/ab0bc1>, <https://doi.org/10.1021/nl504276u>). Note that these previous works have been overlooked in the manuscript and might be considered as a previous foundation of this work. The main novelty of the present method is to exploit the silicon oxide crust on the surface of PDMS (when subjected to O₂ plasma) to engineer the period of the produced ripples. This is very interesting as 1L-WSe₂ should have ripples with a very small period, hampering the use of optical techniques to study them.

After the fabrication of the ripples the authors employed a homebuild confocal system that allows to pump and probe at different sample locations. They used it to test if funnelling effect is present on these samples, finding a very strong effect. This is very relevant as recent works using nanoindentation claimed that funneling was not a relevant mechanism at room temperature. Then the authors finished the manuscript proposing theoretically a device exploiting strain-engineered exciton funneling. The manuscript is well-written, very comprehensive datasets have been acquired, analysed and presented and the quality of the presented figures is very good. I believe that the manuscript will be of the interest of the community working on strain engineering on 2D materials. I, however, have serious doubts that Nature Electronics will be the most appropriated journal as it seems to me that a more optics-focused journal would be more suitable. In view of this I would recommend to send the manuscript to Nature Photonics.

I have further comments that I think would require clarification before re-submission:

Our Response:

We are grateful to the referee for his/her overall positive evaluation on our manuscript. And we thank the referee for bringing relevant literature to our attention. We hope that the referee would be supportive of our manuscript transfer to *Nature Communications* that has a broad scope. In addition, the previous works regarding the fabrication of rippled structures have been added to the revised list of references.

1) Regarding the title. I found the title misleading. First, because the work only deals with one 2D material and not with a broad family of them. Second, because the authors used the term straintronics but no electronic device was build at all. The whole work is an optical spectroscopy work. It is true that there is transport of excitons but honestly no device has been fabricated within this work. It seems to me overclaiming. Actually, I think that the manuscript reads very well until the last discussion about the theoretical proposal for the router. I strongly suggest to move all that discussion to the supporting information. It is interesting but it shifts the focus from the main messages dramatically and makes the reader have that feeling of overclaiming I discussed above.

Our Response:

We thank the referee for the important comments. We agree with the reviewer that our manuscript describes monolayer WSe₂, and thus revised our title as “Strained two-dimensional tungsten diselenide toward straintronics”.

As we discussed in the earlier part of the response letter, straintronics is one of emerging fields in which mechanical strain is used to modulate band structure and resulting electric, optical, magnetic, and other properties for implementation of new information devices and technologies²¹. In particular, exciton is an electron-hole pair bound by Coulomb interaction, and excitons in 2D TMDs can convey information with various types of excitonic characteristics, such as spin and valley, at room temperature as a form of strongly bound bosonic quasiparticle. Thus, controlling exciton for information transport differs from purely photonic devices in which photon is directly modulated by electrical modulator⁶⁴. Although our study focuses on the optical input and optical output using pump-probe PL measurements, exciton generation and signal detection can be carried out by combining with electrical components, such as light emitting tunneling transistor^{65,66}. Still the essential part of exciton transport and information transfer is governed by local strain and strain gradient in our device. In this regard, we believe our exciton routing model aimed at describing exciton transport tuned by strain is a critical first step toward straintronic device demonstration. However, to reflect referee’s comments, we have shortened the exciton router section of our manuscript, moved detailed discussions to SI (Supplementary Text 6), and rearranged the manuscript and Fig. 4 to place a primary focus on the exciton transport simulation. In addition, we made additional efforts to clarify in our manuscript that our discussions on straintronic device model is theoretical in nature.

Page 14 in the manuscript: ~~Modeling of straintronic exciton router~~exciton-based straintronics. We carried out simulation studies of exciton funneling in terms of funneling length and strain beyond our experimental observations. ...

Pages 15-16 in the manuscript: ~~We have demonstrated uniaxial strain induced exciton transport in steady state pumping and probing. On the other hand,~~ To further explore modeling of straintronic device, we propose a theoretical model of straintronic exciton router on the basis of our experimental results and analysis on strained WSe₂. Straintronic logic device may require controllable exciton routing ~~and transient exciton transport for ultrafast signal processing and logic operation,~~ which converts input light to excitons, programmably manipulates exciton flux to specific direction, and outputs signal as a form of photon that is created by recombined excitons^{1,2}. In this regard, we build a theoretical model of straintronic exciton router which is functional at GHz frequency signal processing on the basis of our experimental results and analysis on strained WSe₂. In this ~~our~~ model, WSe₂ is conformally deformed as checkerboard-shape and exerted with biaxial tensile/compressive strain at the peak/valley of the structure (Fig. 4d-e). We intend to control exciton transport direction as a function of reconfigurable local strain (Fig. 4c). For instance, stretching in x direction causes elimination of x direction local strain component ($\epsilon_{xx}=0$), resulting in exciton flux toward y direction, denoted as (0,1) coordinate. The exciton transport directions and consequent emission port are determined by reconfigurable local strain gradient by applying external strain (detailed description of the routing model in Supplementary Text 6). The normalized exciton density at $t = 0$ ns and $t = 2$ ns is simulated and plotted in Fig. 4d and Fig. 4e-g, respectively. Excitation at the valley of the structure creates high energy excitons at $t = 0$ ns, which flow toward either (1,0), (0,1), or (1,1) direction based on the local strain distribution. Specifically, biaxial local strain causes exciton drift toward (1,1) direction. By changing local strain (e.g., stretching) in x- and y- directions we can induce exciton drift to (0,1) and (1,0) direction, respectively. The rate of exciton transport can be quantified by exponential

time constant at the rising edge (τ_e) and 10/90 rise time ($\tau_{10/90}$)³ for which input excitation is converted to maximum output emission at the target position. It is noted that the strain-induced exciton transport can be fast enough to achieve a response of output emission photon in nanosecond timescale (i.e., GHz transmission frequency) (Supplementary Fig. 14), ~~which revealed that straintronic device is functional at GHz frequencies (Supplementary Fig. 12).~~ We attribute the rapid exciton transport to ~~reduced scattering of charge neutral exciton~~ high mobility of charge-neutral excitons ($300\text{-}576\text{ cm}^2/\text{Vs}$)^{4,5} compared with low diffusivity and mobility of interlayer excitons in 2D vdW heterostructures due to trapping in moiré potential⁶⁻⁸ as well as smooth energy gradient along the exciton transport path.

2) The mechanical model used to calculate the strain out of the curvature radius of the ripples is not accurate. 1L WSe₂ is mainly a membrane, not a plate. And the model that they are using is only valid for plates whose mechanics are dominated by bending rigidity. In the case of a single-layer WSe₂, the bending rigidity is negligible in comparison to the pre-tension. So, calculating the strain through the curvature radius is not correct. That calculation can be OK for the SiO₂ skin layer (being ~12 nm thick and with a large young's modulus its bending rigidity has to be sizeable). Then the authors can claim that they calculate the strain on the SiO₂ layer and that they assume perfect strain transfer to the WSe₂ as a first approximation.

Our Response:

We thank the referee for this comment. As the referee pointed out, plate theories in continuum mechanics are not suited to estimate strain applied on atomically-thin membranes due to atomic thinness and low bending rigidity. We recognized such limitation and stated our assumptions of strain estimation using linear elasticity theory in Supplementary Text 1 of our original SI: (1) wrinkles follow periodic sinusoidal shape, (2) the effect of PMMA is negligible due to relatively lower adhesion force to WSe₂, (3) strong adhesion at both silica/PDMS and silica/WSe₂ interfaces. The third assumption corresponds to the perfect strain transfer between silica skin layer and WSe₂. This assumption requires further justification by analyzing interfacial adhesion and excluding slippage/delamination at the interface, which is beyond the scope of this paper. Nevertheless, perfect strain transfer has been assumed in a wide range of strain analysis of 2D materials, particularly when it comes to 2D materials supported on a substrate. One example is to apply uniform tensile strain on 2D materials by placing 2D flake on thick polymer substrates (e.g., polycarbonate, polyethylene terephthalate glycol, polyvinyl alcohol) and conformally bending 2D/polymer substrate. Uniform strain was calculated with simple bending model ($\epsilon = t / 2R$, where t is thickness of 2D/polymer and R is radius of curvature). This model assumes (1) complete strain transfer between polymer substrate and supported 2D material, (2) strain-free state of as-fabricated flat sample, and (3) no lateral strain gradient in 2D material. This method has been widely adopted for various types of 2D materials (graphene, MoS₂, MoSe₂, WSe₂, WS₂) to measure strain-tunable properties of 2D materials⁶⁷⁻⁷³. Particularly, the adhesion between SiO₂ and TMDs used in our experiment is stronger than normal van der Waals interaction due to increased interfacial bonding between chalcogen and oxygen atoms⁷⁴. Thus, our assumption of strain transfer for linear elasticity theory is not deviated from previous reports of strain estimation, while more rigorous analysis on the local strain is necessary in the future. To give readers a clearer understanding of this, we added additional discussions in the Supplementary Information (Supplementary Text 1).

Pages 4-5 in the Supplementary Information: Since the plate theories in continuum mechanics are not suited to estimate strain applied on atomically-thin membranes due to atomic thinness and low bending rigidity, we focus on the maximum strain applied at the apex/valley of silica skin layer and assume

perfect strain transfer between silica skin layer and WSe₂ (assumption (3)). This assumption requires further justification by analyzing interfacial adhesion and excluding slippage/delamination at the interface, which is beyond the scope of this paper. Nevertheless, perfect strain transfer has been assumed in a wide range of strain analysis of 2D materials, particularly when it comes to 2D materials supported on a substrate. One example is to apply uniform tensile strain on 2D materials by placing 2D flake on thick polymer substrates (e.g., polycarbonate, polyethylene terephthalate glycol, polyvinyl alcohol) and conformally bending 2D/polymer substrate. Uniform strain was calculated with simple bending model ($\epsilon = t / 2R$, where t is thickness of 2D/polymer and R is radius of curvature). This model assumes (1) complete strain transfer between polymer and supported 2D material, (2) strain-free state of as-fabricated flat sample, and (3) no lateral strain gradient in 2D material. This method has been widely adopted for various types of 2D materials (graphene, MoS₂, MoSe₂, WSe₂, WS₂) to measure strain-tunable properties of 2D materials⁶⁷⁻⁷³.

3) Page 6, line 116: "The local strain value here exceeds those previously achieved by strain engineering of monolayer TMDs^{17,21,26,35}". But in ref [<https://doi.org/10.1021/acs.nanolett.6b02615>] local strains up to 5.6% are reported.

Our Response:

We thank the referee for this valuable comment. In the paper the reviewer mentioned [*Nano Lett.*, 16, 5836 (2016)], the high strain was applied on monolayer MoS₂ via bulging approach. However, the strain was uniform over the bulged MoS₂ area with a diameter of 8 μ m, and thus, the strain gradient effect (PL intensity enhancement) was not observed. To emphasize this difference in our work, we have revised the expression "strain engineering of monolayer TMDs" into "*local* strain engineering of monolayer TMDs" and cited the paper which the reviewer referred to.

*Page 7 in the manuscript: The local strain value here exceeds those previously achieved by **local** strain engineering of monolayer TMDs^{4,14,46,71}*

Reference

1. High, A. A., Novitskaya, E. E., Butov, L. V., Hanson, M. & Gossard, A. C. Control of exciton fluxes in an excitonic integrated circuit. *Science* **321**, 229–231 (2008).
2. Liu, Y. *et al.* Electrically controllable router of interlayer excitons. *Sci. Adv.* **6**, eaba1830 (2020).
3. High, A. A., Hammack, A. T., Butov, L. V., Hanson, M. & Gossard, A. C. Exciton optoelectronic transistor. *Opt. Lett.* **32**, 2466 (2007).
4. Cordovilla Leon, D. F., Li, Z., Jang, S. W., Cheng, C. H. & Deotare, P. B. Exciton transport in strained monolayer WSe₂. *Appl. Phys. Lett.* **113**, 252101 (2018).
5. Cadiz, F. *et al.* Exciton diffusion in WSe₂ monolayers embedded in a van der Waals heterostructure. *Appl. Phys. Lett.* **112**, 152106 (2018).
6. Li, Z. *et al.* Interlayer Exciton Transport in MoSe₂/WSe₂ Heterostructures. *ACS Nano* **15**, 1539–1547 (2021).
7. Miao, S. *et al.* Strong interaction between interlayer excitons and correlated electrons in WSe₂/WS₂ moiré superlattice. *Nat. Commun.* **12**, 3608 (2021).
8. Yuan, L. *et al.* Twist-angle-dependent interlayer exciton diffusion in WS₂–WSe₂ heterobilayers. *Nat. Mater.* **19**, 617–623 (2020).
9. Fandan, R. *et al.* Dynamic Local Strain in Graphene Generated by Surface Acoustic Waves. *Nano Lett.* **20**, 402–409 (2020).
10. Pellegrino, L., Khodaparast, S. & Cabral, J. T. Orthogonal wave superposition of wrinkled, plasma-oxidised, polydimethylsiloxane surfaces. *Soft Matter* **16**, 595–603 (2020).
11. Cai, S., Breid, D., Crosby, A. J., Suo, Z. & Hutchinson, J. W. Periodic patterns and energy states of buckled films on compliant substrates. *J. Mech. Phys. Solids* **59**, 1094–1114 (2011).
12. Chen, X. & Hutchinson, J. W. Herringbone buckling patterns of compressed thin films on compliant substrates. *J. Appl. Mech. Trans. ASME* **71**, 597–603 (2004).
13. Chen, X. & Hutchinson, J. W. A family of herringbone patterns in thin films. *Scr. Mater.* **50**, 797–801 (2004).
14. Li, H. *et al.* Optoelectronic crystal of artificial atoms in strain-textured molybdenum disulphide. *Nat. Commun.* **6**, 7381 (2015).
15. Aas, S. & Bulutay, C. Strain dependence of photoluminescence and circular dichroism in transition metal dichalcogenides: a k · p analysis. *Opt. Express* **26**, 28672 (2018).
16. Rowland, R. J. *Principles of solid mechanics*. (CRC Press, 2019).
17. Reddy, J. N. *Theory and analysis of elastic plates and shells*. (CRC Press, 2006).
18. Mitchell, N. P. *et al.* Conforming nanoparticle sheets to surfaces with Gaussian curvature. *Soft Matter* **14**, 9107–9117 (2018).
19. Liu, X. & Guo, W. Shear strain tunable exciton dynamics in two-dimensional semiconductors. *Phys. Rev. B* **99**, 035401 (2019).
20. Unuchek, D. *et al.* Room-temperature electrical control of exciton flux in a van der Waals heterostructure. *Nature* **560**, 340–344 (2018).
21. Bukharaev, A. A., Zvezdin, A. K., Pyatakov, A. P. & Fetisov, Y. K. Straintronics: a new trend in micro- and nanoelectronics and material science. *Uspekhi Fiz. Nauk* **188**, 1288–1330 (2018).
22. Gourley, P. L. & Wolfe, J. P. Spatial condensation of strain-confined excitons and excitonic molecules into an electron-hole liquid in silicon. *Phys. Rev. Lett.* **40**, 526–530 (1978).
23. Koperski, M. *et al.* Single photon emitters in exfoliated WSe₂ structures. *Nat. Nanotechnol.* **10**, 503–506 (2015).

24. Chakraborty, C., Kinnischtzke, L., Goodfellow, K. M., Beams, R. & Vamivakas, A. N. Voltage-controlled quantum light from an atomically thin semiconductor. *Nat. Nanotechnol.* **10**, 507–511 (2015).
25. He, Y. M. *et al.* Single quantum emitters in monolayer semiconductors. *Nat. Nanotechnol.* **10**, 497–502 (2015).
26. Guinea, F., Katsnelson, M. I. & Geim, A. K. Energy gaps and a zero-field quantum hall effect in graphene by strain engineering. *Nat. Phys.* **6**, 30–33 (2010).
27. Chang, Y., Albash, T. & Haas, S. Quantum Hall states in graphene from strain-induced nonuniform magnetic fields. *Phys. Rev. B - Condens. Matter Mater. Phys.* **86**, 125402 (2012).
28. Harats, M. G., Kirchhof, J. N., Qiao, M., Greben, K. & Bolotin, K. I. Dynamics and efficient conversion of excitons to trions in non-uniformly strained monolayer WS₂. *Nat. Photonics* **14**, 324–329 (2020).
29. Miller, D. A. B. Rationale and challenges for optical interconnects to electronic chips. *Proc. IEEE* **88**, 728–749 (2000).
30. O'Brien, J. L., Furusawa, A. & Vučković, J. Photonic quantum technologies. *Nat. Photonics* **3**, 687–695 (2009).
31. Mueller, T. & Malic, E. Exciton physics and device application of two-dimensional transition metal dichalcogenide semiconductors. *npj 2D Mater. Appl.* **2**, 29 (2018).
32. He, K. *et al.* Tightly bound excitons in monolayer WSe₂. *Phys. Rev. Lett.* **113**, 026803 (2014).
33. Schaibley, J. R. *et al.* Valleytronics in 2D materials. *Nat. Rev. Mater.* **1**, 16055 (2016).
34. Unuchek, D. *et al.* Valley-polarized exciton currents in a van der Waals heterostructure. *Nat. Nanotechnol.* **14**, 1104 (2019).
35. Koperski, M. *et al.* Single photon emitters in exfoliated WSe₂ structures. *Nat. Nanotechnol.* **10**, 503–506 (2015).
36. Palacios-Berraquero, C. *et al.* Large-scale quantum-emitter arrays in atomically thin semiconductors. *Nat. Commun.* **8**, 15093 (2017).
37. Yoshioka, K., Chae, E. & Kuwata-Gonokami, M. Transition to a Bose-Einstein condensate and relaxation explosion of excitons at sub-Kelvin temperatures. *Nat. Commun.* **2**, 328 (2011).
38. Fu, X. *et al.* Tailoring exciton dynamics by elastic strain-gradient in semiconductors. *Adv. Mater.* **26**, 2572–2579 (2014).
39. Stier, A. V., Wilson, N. P., Clark, G., Xu, X. & Crooker, S. A. Probing the Influence of Dielectric Environment on Excitons in Monolayer WSe₂: Insight from High Magnetic Fields. *Nano Lett.* **16**, 7054–7060 (2016).
40. Malyi, O. I. *et al.* Volume dependence of the dielectric properties of amorphous SiO₂. *Phys. Chem. Chem. Phys.* **18**, 7483–7489 (2016).
41. Hsu, W. T. *et al.* Dielectric impact on exciton binding energy and quasiparticle bandgap in monolayer WS₂ and WSe₂. *2D Mater.* **6**, 025028 (2019).
42. Zeng, X. & Jiang, H. Strain effect of the dielectric constant in silicon dioxide. *J. Microelectromechanical Syst.* **20**, 353–354 (2011).
43. Moon, H. *et al.* Dynamic Exciton Funneling by Local Strain Control in a Monolayer Semiconductor. *Nano Lett.* **20**, 6791–6797 (2020).
44. Falin, A. *et al.* Mechanical properties of atomically thin boron nitride and the role of interlayer interactions. *Nat. Commun.* **8**, 15815 (2017).
45. Chang, J., Toga, K. B., Paulsen, J. D., Menon, N. & Russell, T. P. Thickness Dependence of the

- Young's Modulus of Polymer Thin Films. *Macromolecules* **51**, 6764–6770 (2018).
46. Niehues, I. *et al.* Strain Control of Exciton-Phonon Coupling in Atomically Thin Semiconductors. *Nano Lett.* **18**, 1751–1757 (2018).
 47. Aslan, O. B., Deng, M. & Heinz, T. F. Strain tuning of excitons in monolayer WSe₂. *Phys. Rev. B* **98**, 115308 (2018).
 48. Arora, A. *et al.* Excitonic resonances in thin films of WSe₂: From monolayer to bulk material. *Nanoscale* **7**, 10421–10429 (2015).
 49. You, Y. *et al.* Observation of biexcitons in monolayer WSe₂. *Nat. Phys.* **11**, 477–481 (2015).
 50. Cai, T. *et al.* Coupling Emission from Single Localized Defects in Two-Dimensional Semiconductor to Surface Plasmon Polaritons. *Nano Lett.* **17**, 6564–6568 (2017).
 51. Oliva-Leyva, M. & De La Cruz, G. G. Unveiling optical in-plane anisotropy of 2D materials from oblique incidence of light. *J. Phys. Condens. Matter* **31**, 335701 (2019).
 52. Matthes, L., Pulci, O. & Bechstedt, F. Influence of out-of-plane response on optical properties of two-dimensional materials: First principles approach. *Phys. Rev. B* **94**, 205408 (2016).
 53. Majérus, B., Dremetsika, E., Lobet, M., Henrard, L. & Kockaert, P. Electrodynamics of two-dimensional materials: Role of anisotropy. *Phys. Rev. B* **98**, 125419 (2018).
 54. Wang, G. *et al.* In-Plane Propagation of Light in Transition Metal Dichalcogenide Monolayers: Optical Selection Rules. *Phys. Rev. Lett.* **119**, 047401 (2017).
 55. Park, K. D., Jiang, T., Clark, G., Xu, X. & Raschke, M. B. Radiative control of dark excitons at room temperature by nano-optical antenna-tip Purcell effect. *Nat. Nanotechnol.* **13**, 59–64 (2018).
 56. So, J. P. *et al.* Polarization Control of Deterministic Single-Photon Emitters in Monolayer WSe₂. *Nano Lett.* **21**, 1546–1554 (2021).
 57. Yang, S. *et al.* Tuning the optical, magnetic, and electrical properties of ReSe₂ by nanoscale strain engineering. *Nano Lett.* **15**, 1660–1666 (2015).
 58. Castellanos-Gomez, A. *et al.* Local strain engineering in atomically thin MoS₂. *Nano Lett.* **13**, 5361–5366 (2013).
 59. Wang, G. *et al.* Valley dynamics probed through charged and neutral exciton emission in monolayer WSe₂. *Phys. Rev. B - Condens. Matter Mater. Phys.* **90**, 075413 (2014).
 60. Palummo, M., Bernardi, M. & Grossman, J. C. Exciton radiative lifetimes in two-dimensional transition metal dichalcogenides. *Nano Lett.* **15**, 2794–2800 (2015).
 61. Zhang, X. X., You, Y., Zhao, S. Y. F. & Heinz, T. F. Experimental Evidence for Dark Excitons in Monolayer WSe₂. *Phys. Rev. Lett.* **115**, 257403 (2015).
 62. Mouri, S. *et al.* Nonlinear photoluminescence in atomically thin layered WSe₂ arising from diffusion-assisted exciton-exciton annihilation. *Phys. Rev. B - Condens. Matter Mater. Phys.* **90**, 155449 (2014).
 63. Yan, T., Qiao, X., Liu, X., Tan, P. & Zhang, X. Photoluminescence properties and exciton dynamics in monolayer WSe₂. *Appl. Phys. Lett.* **105**, 101901 (2014).
 64. Grosso, G. *et al.* Excitonic switches operating at around 100K. *Nat. Photonics* **3**, 577–580 (2009).
 65. Wang, Z. *et al.* Evidence of high-temperature exciton condensation in two-dimensional atomic double layers. *Nature* **574**, 76–80 (2019).
 66. Withers, F. *et al.* WSe₂ Light-Emitting Tunneling Transistors with Enhanced Brightness at Room Temperature. *Nano Lett.* **15**, 8223–8228 (2015).
 67. Hu, G. *et al.* Controlling the Dirac point voltage of graphene by mechanically bending the ferroelectric gate of a graphene field effect transistor. *Mater. Horizons* **6**, 302–310 (2019).

68. Mohiuddin, T. M. G. *et al.* Uniaxial strain in graphene by Raman spectroscopy: G peak splitting, Grüneisen parameters, and sample orientation. *Phys. Rev. B - Condens. Matter Mater. Phys.* **79**, 205433 (2009).
69. Conley, H. J. *et al.* Bandgap engineering of strained monolayer and bilayer MoS₂. *Nano Lett.* **13**, 3626–3630 (2013).
70. Desai, S. B. *et al.* Strain-induced indirect to direct bandgap transition in multilayer WSe₂. *Nano Lett.* **14**, 4592–4597 (2014).
71. Schmidt, R. *et al.* Reversible uniaxial strain tuning in atomically thin WSe₂. *2D Mater.* **3**, 021011 (2016).
72. Island, J. O. *et al.* Precise and reversible band gap tuning in single-layer MoSe₂ by uniaxial strain. *Nanoscale* **8**, 2589–2593 (2016).
73. Li, Z. *et al.* Efficient strain modulation of 2D materials via polymer encapsulation. *Nat. Commun.* **11**, 1151 (2020).
74. Torres, J., Zhu, Y., Liu, P., Lim, S. C. & Yun, M. Adhesion Energies of 2D Graphene and MoS₂ to Silicon and Metal Substrates. *Phys. Status Solidi Appl. Mater. Sci.* **215**, 1700512 (2018).

RESPONSE TO REVIEWERS' COMMENTS

Reviewer #2 (Remarks to the Author):

I greatly appreciate the authors' efforts to address my previous comments/questions, along with those from other reviewers. The authors have gone to great lengths doing so and have addressed most of my questions.

However, I still have one major concern. The fact that the low-temperature PL of the flat region of WSe₂ in Fig. S15 is troubling. Not only the PL peak is at lower energy than the A exciton, but it is also extremely weak, and the line shape is really weird. It suggests extremely poor quality of the WSe₂ at the flat region and significant contribution of defects. This interpretation would be consistent with the very long lifetime the authors observed at room temperature (Table R1). Is it possible that all the observations reported in this work are simply the strain modulation of the defect states in WSe₂? Where would all these defects come from? Over the last few years, there have been significant developments in WSe₂ crystal growth, and the quality could be very high. Could the authors prepare one monolayer WSe₂ with PMMA capping layer but no strain and then check the PL at low temperature? How would this PL compare with previous reports, and could authors explain the low-temperature PL observed in S15?

Our Response:

We appreciate the referee for positive evaluation of our response and revised manuscript. To address the reviewer's concern, we carried out low-temperature PL measurement on unstrained WSe₂ with and without a PMMA capping layer (new Figure S15a-d). As temperature decreases, PL emission of A exciton was blueshifted and attenuated, regardless of the presence of a PMMA capping layer. In addition, we observed multiple PL emission peaks emerging below 100 K. Relatively stronger emission of new PL peaks compared to that of bright A exciton at low temperature has been also observed in previous reports¹⁻⁴, which is attributed to reduced thermalization and increased probability to form low-energy, weakly-bound excitonic components such as trion (1.69 eV)¹, biexciton (1.68 eV)², and/or localized/bound excitons (<1.68 eV)^{3,4}. It should be noted that the weak PL emission in our previous response is due to low excitation power, rather than defects. Moreover, the WSe₂ crystal we used was grown by chemical vapor transport (CVT) method with high purity (>99.995%, HQ Graphene), and also has been widely utilized in 2D materials research field⁶⁻⁹. Thus, we can conclude that our observation of exciton transport at room temperature originates from strain-induced funneling of thermalized A exciton, rather than defects or localized emissions.

To respond to the reviewer's comment, we have revised the Supplementary Information and Supplementary Fig. 15.

Pages 6-7 in the Supplementary Information: To further elucidate the asymmetry of PL emission, we performed low-temperature PL measurement ($T=15\text{K}$) of the flat and wrinkled WSe₂ prepared using the same fabrication condition (Supplementary Fig. S15). The unstrained flat WSe₂ showed relatively weak and blueshifted PL emission at 740 nm (1.675 eV) of A exciton at low temperature (Supplementary Fig. 15a-d). In addition, we observed several peaks emerged at low temperature, which may be ascribed to reduced thermalization and increased probability to form low-energy, weakly-bound excitonic components such as trion (1.69 eV)¹, biexciton (1.68 eV)², and/or localized/bound excitons (<1.68 eV)^{3,4}. These trends of temperature-dependent PL emission were not affected by presence of a PMMA encapsulating layer, indicating negligible doping effect of polymer capping. The observed PL emission corresponds to localized/bound excitons rather than charged excitons (1.70 eV) or neutral excitons (1.72 eV)³. It is also close to PL emission of biexcitons (1.68 eV), while biexciton

plays less significant roles in PL emission at room temperature due to thermal dissociation of weakly bound biexciton². On the other hand, we observed sharp, strong, and more symmetric PL emission along apex lines of the wrinkled WSe₂ at low temperature (Supplementary Fig. 15e, f). The observed PL emission peaks in wrinkled WSe₂ may be attributed to excitons confined in locally strained area⁴ with reduced phonon scattering at low temperature. However, such sharp emission of localized exciton was observed only at low temperature (Supplementary Fig. 15f) because of thermal detrapping of bound excitons at elevated temperature. It should be noted that the PL emission of wrinkled WSe₂ measured at low temperature does not match with emission of different excitonic components. These results suggest that the asymmetric PL emission of locally strained WSe₂ at room temperature may not result from other excitonic components, but from strain gradient induced effects on bright A exciton such as energy gradient within the excitation area and/or exciton funneling.

New Figure S15. PL emission of unstrained and strained WSe₂ at varying temperature. **a**, Optical microscope image of unstrained flat WSe₂. **b**, The same WSe₂ with PMMA thin film coated on top. **c-d**, PL emission spectra of unstrained WSe₂ without PMMA coating (**c**) and with PMMA coating (**d**). **e**, Fluorescence intensity map of wrinkled WSe₂, showing localized strong PL emission along the apex line. The scale bar is 5 μm . **f**, PL spectra of wrinkled WSe₂ at localized emission points in (**e**) at T = 15 K.

Reviewer #3 (Remarks to the Author):

The authors have addressed most of the points raised in my referee report very thoroughly. However, I believe that some of the points have not been addressed successfully:

1) Point #1. Regarding the discussion about the use of straintronics in the title, I still believe that its use is somewhat overclaiming. In my view what they are mostly doing is strain engineering not straintronics, straintronics requires the fabrication of a device exploiting strain tunability.

Our Response:

We appreciate the referee for positive evaluation of our response and revised manuscript. We have revised our title to “Strained two-dimensional tungsten diselenide ~~toward straintronics~~ **for mechanically tunable exciton transport**”

2) Point #3. The authors proposed the change: "...The local strain value here exceeds those previously achieved by ****local**** strain engineering of monolayer TMDs^{4,14,46,71}. But the strain in [Nano Lett., 16, 5836 (2016)] is also local (see the supporting information of that manuscript, for example). So their claim about the larger reported local strain is not accurate.

Our Response:

We thank the referee for the comment. We have revised our manuscript to convey clear information on this.

*Page 7 in the manuscript: The local strain value here exceeds those previously achieved by ~~local~~ **underlying substrate-induced** strain engineering of monolayer TMDs¹⁰⁻¹³*

Reference

1. Wang, G. *et al.* In-Plane Propagation of Light in Transition Metal Dichalcogenide Monolayers: Optical Selection Rules. *Phys. Rev. Lett.* **119**, 047401 (2017).
2. You, Y. *et al.* Observation of biexcitons in monolayer WSe₂. *Nat. Phys.* **11**, 477–481 (2015).
3. Arora, A. *et al.* Excitonic resonances in thin films of WSe₂: From monolayer to bulk material. *Nanoscale* **7**, 10421–10429 (2015).
4. Cai, T. *et al.* Coupling Emission from Single Localized Defects in Two-Dimensional Semiconductor to Surface Plasmon Polaritons. *Nano Lett.* **17**, 6564–6568 (2017).
5. Mueller, T. & Malic, E. Exciton physics and device application of two-dimensional transition metal dichalcogenide semiconductors. *npj 2D Mater. Appl.* **2**, 29 (2018).
6. Fülöp, B. *et al.* Boosting proximity spin–orbit coupling in graphene/WSe₂ heterostructures via hydrostatic pressure. *npj 2D Mater. Appl.* **5**, 82 (2021).
7. Movva, H. C. P. *et al.* High-Mobility Holes in Dual-Gated WSe₂ Field-Effect Transistor. *ACS Nano* **9**, 10402 (2015).
8. Fallahazad, B. *et al.* Shubnikov-de Haas Oscillations of High-Mobility Holes in Monolayer and Bilayer WSe₂: Landau Level Degeneracy, Effective Mass, and Negative Compressibility. *Phys. Rev. Lett.* **116**, 086601 (2016).
9. Bertoni, R. *et al.* Generation and Evolution of Spin-, Valley-, and Layer-Polarized Excited Carriers in Inversion-Symmetric WSe₂. *Phys. Rev. Lett.* **117**, 277201 (2016).
10. Li, H. *et al.* Optoelectronic crystal of artificial atoms in strain-textured molybdenum disulphide. *Nat. Commun.* **6**, 7381 (2015).
11. Niehues, I. *et al.* Strain Control of Exciton-Phonon Coupling in Atomically Thin Semiconductors. *Nano Lett.* **18**, 1751–1757 (2018).
12. Cordovilla Leon, D. F., Li, Z., Jang, S. W., Cheng, C. H. & Deotare, P. B. Exciton transport in strained monolayer WSe₂. *Appl. Phys. Lett.* **113**, 252101 (2018).
13. Schmidt, R. *et al.* Reversible uniaxial strain tuning in atomically thin WSe₂. *2D Mater.* **3**, 021011 (2016).

RESPONSE TO REVIEWERS' COMMENTS

Comment. The authors have obtained low temperature PL of unstrained WSe₂, with and without PMMA. I found that the analysis and discussion of the data did not support the authors' claim. Here are my comments:

Comment 1. The authors should plot the low temperature PL from unstrained WSe₂ without and with PMMA on the same panel (panels c and d). With the current layout, it is hard to see the difference. But even with the current layout, it is clear that the WSe₂ with PMMA has more significant PL peaks at the lower energy. This strongly contradicts the authors' claim "These trends of temperature-dependent PL emission were not affected by presence of a PMMA encapsulating layer, indicating negligible doping effect of polymer capping."

Our response. As the reviewer suggested, we plotted the low temperature PL from unstrained WSe₂ with and without PMMA on the same panel (new Fig. S15a and new Fig. S16). This graph shows that PL peaks with energies below 1.65 eV exhibit minor differences at a temperature below 100 K depending on the presence of PMMA, as the reviewer pointed out. However, the trends of temperature-dependent PL emission are not affected by presence of PMMA in the regions of neutral exciton (1.74 eV), trion (1.71 eV), biexciton (1.68 eV), and part of localized excitons (1.66-1.68 eV).¹⁻³ In addition, the effect of strain was manifested by substantial PL peak shift at both low and ambient temperatures, making it distinguishable from PMMA-induced effects.

To respond to the reviewer's comment, we revised the discussion as follows:

Page 6 in the Supplementary Information: To further elucidate the asymmetry of PL emission, we performed low-temperature PL measurement of the flat and wrinkled WSe₂ prepared using the same fabrication condition (Supplementary Fig. 15-17). The unstrained flat WSe₂ showed relatively weak and blueshifted PL emission of A exciton at low temperature compared to that measured at room temperature ($E = 1.74$ eV) (Supplementary Fig. 15a-d). In addition, we observed several peaks emerged at low temperature, which may be ascribed to reduced thermalization and increased probability to form low-energy, weakly-bound excitonic components such as trion (1.71 eV)¹², biexciton (1.68 eV)¹³, and/or localized/bound excitons (<1.68 eV)^{14,15}. These trends of temperature-dependent PL emission were not affected in the regions of neutral exciton and trion with presence of a PMMA encapsulating layer, indicating minimal effect of polymer capping. ~~by presence of a PMMA encapsulating layer, indicating negligible doping effect of polymer capping.~~ We note that PL peaks with energies below 1.65 eV exhibit minor differences at

less than 100 K depending on the presence of PMMA, whereas such differences tend to disappear as increasing temperature (Supplementary Fig. 16).

New Fig. S15. PL emission of unstrained/strained WSe₂ with and without PMMA thin film at varying temperature. a, PL emission spectra measured at 15 K. **b,** PL emission spectra measured at 298 K. Neutral exciton (X⁰) and trion (T) were indexed in the plot. For the unstrained PL emission, we used the same monolayer WSe₂ flake with and without PMMA coating at both temperatures, while we employed different strained WSe₂ flakes due to wrinkle formation.

New Fig. S16. PL emission of unstrained WSe₂ with and without PMMA thin film at varying temperature. PL measurement was carried out at (a) 4 K, (b) 10 K, (c) 20 K, (d) 40 K, (e) 70 K, (f) 100 K, (g) 150 K, (h) 200 K, and (i) 298 K.

Comment 2. The authors mentioned a couple of possibilities to explain the low temperature PL. However, it was not clear which one was which. The cited possibilities are in eV while the PL was plotted as a function of wavelength, and the way it is plotted also makes it difficult to understand. Could the authors change the x-axis to the unit of eV and try to identify the exciton peak, biexciton, localized exciton, etc.? The figure also needs to be made significantly larger to show the low temperature PL clearly.

Our response. As the reviewer suggested, we changed the x-axis to the unit of eV, and made the figure considerably larger to clearly show the low temperature PL. In addition, we identified the exciton peaks which were shown in our measurement (new Fig. S15).

Comment 3. Is the localized exciton state due to defects? If so, how does that affect the current interpretation? The biexciton peak can potentially be separated from the exciton through an excitation power dependence study.

Our response. The optically bright localized excitons are possibly related to disorder, scattering or crystal imperfections, while their nature is not fully understood^{4,5}. However, even in this case, our current interpretation is not affected because the sharp emission of localized exciton was observed only at low temperature, as we already discussed in the Supplementary Information. In addition, PL emission from WSe₂ biexciton (1.68 eV) is reported to quickly decay at temperature above 40 K due to low binding energy of biexciton (30-52 meV) (e.g., Fig. 2c in Ref. 3). Therefore, we believe the localized exciton and biexciton are less relevant to our main claim of observing exciton transport at room temperature.

Comment 4. The authors should plot the 15 K data of the unstrained WSe₂, without and with PMMA, in the same panel of the pane (f). What is the nature of the enhanced PL in panel (f)? Are they free excitons, localized excitons, or other quasiparticles?

Our response. As the reviewer suggested, for the direct comparison, we plotted the spectra of the unstrained WSe₂ at 15 K and 298 K, with and without PMMA in the same panel of strained WSe₂ (new Fig. S15). The enhanced PL of strained WSe₂ in Fig. S17 is originated from a dense ensemble of localized excitons by exciton funneling at the wrinkle. The enhanced PL in such strained structures has also been observed in other reports.⁶⁻⁸

To respond to the reviewer's comment, we revised the discussion as follows:

*Pages 6-7 in the Supplementary Information: On the other hand, we observed ~~sharp, strong, and more symmetric PL emission~~ **strong PL emission** along apex lines of the wrinkled WSe₂ at low temperature (Supplementary Fig. ~~17~~**15e, f**). The observed PL emission peaks in wrinkled WSe₂ may be attributed to **dense ensemble of localized excitons by exciton funneling at the wrinkle apex** ~~excitons confined in locally strained area~~¹⁵ with reduced phonon scattering at low temperature. However, such sharp emission of localized exciton was observed only at low*

temperature (Supplementary Fig. 15f) because of thermal detrapping of bound excitons at elevated temperature. These results suggest that the asymmetric PL emission of locally strained WSe₂ at room temperature may not result from other excitonic components, but from strain gradient induced effects on bright A exciton such as energy gradient within the excitation area and/or exciton funneling.

Fig. S17. Enhanced PL emission of strained WSe₂ at low temperature. **a**, Fluorescence intensity map of wrinkled WSe₂, showing localized strong PL emission along the apex line. The scale bar is 5 μ m. **b**, PL spectra of wrinkled WSe₂ at localized emission points in (a) at T = 15 K.

Comment 5. The authors claim the weak PL was due to low excitation power. Do the authors have an estimate of the quantum yield? It will be a better way to compare with others' work, especially the high-quality samples. Using widely used commercially available crystals does not guarantee a good sample because the crystal is likely to differ from batch to batch. In addition, the community has found that the sample quality is highly dependent on the preparation procedure.

Our response. We agree with the reviewer that the sample quality might differ from batch to batch and can be highly dependent on the preparation procedure. However, we believe the quantum yield is less relevant to our main claim of observing exciton transport at room temperature. In particular,

we believe that our sample quality is good enough to show reliable and reproducible measurements of strain-induced exciton funneling at room temperature as shown in Supplementary Fig. S3. Furthermore, the weak PL in the flat area (Fig. S15), due to low excitation power, was also reported in other previous works. For example, Fig. 2g in Ref. 6 and Fig. 2c in Ref. 8 also showed weak PL intensity. These low-temperature spectra of WSe₂ are consistent with our findings in the present work.

References

1. Arora, A. *et al.* Excitonic resonances in thin films of WSe₂: From monolayer to bulk material. *Nanoscale* **7**, 10421–10429 (2015).
2. Huang, J., Hoang, T. B. & Mikkelsen, M. H. Probing the origin of excitonic states in monolayer WSe₂. *Sci. Rep.* **6**, 22414 (2016).
3. You, Y. *et al.* Observation of biexcitons in monolayer WSe₂. *Nat. Phys.* **11**, 477–481 (2015).
4. Dery, H. & Song, Y. Polarization analysis of excitons in monolayer and bilayer transition-metal dichalcogenides. *Phys. Rev. B - Condens. Matter Mater. Phys.* **92**, 125431 (2015).
5. Slobodeniuk, A. O. & Basko, D. M. Spin-flip processes and radiative decay of dark intravalley excitons in transition metal dichalcogenide monolayers. *2D Mater.* **3**, 035009 (2016).
6. Branny, A., Kumar, S., Proux, R. & Gerardot, B. D. Deterministic strain-induced arrays of quantum emitters in a two-dimensional semiconductor. *Nat. Commun.* **8**, 15053 (2017).
7. Palacios-Berraquero, C. *et al.* Large-scale quantum-emitter arrays in atomically thin semiconductors. *Nat. Commun.* **8**, 15093 (2017).
8. So, J. P. *et al.* Polarization Control of Deterministic Single-Photon Emitters in Monolayer WSe₂. *Nano Lett.* **21**, 1546–1554 (2021).

RESPONSE TO REVIEWERS' COMMENTS

Reviewer #2 (Remarks to the Author):

I appreciate the authors' efforts to reorganize and discuss their low-temperature data. I also want to emphasize that I do appreciate the authors' idea of using strain to manipulate the bandstructure spatially to realize exciton funneling. However, the reply did not fully address my concerns, and it actually raised serious questions about the interpretation of the data. Therefore, I would not recommend the publication of this work in Nature Communications.

My main concern is that the enhanced PL is mostly from defects, affected by the strain, instead of from exciton funneling from the local bandstructure modulation by strain.

I will list my concerns to the reply first and then summarize why these concerns raise my doubts about the data interpretation.

Comment 1: In response to my comment 1, the authors did put PL spectra on the simple figure, which is much easier for comparison. However, the exciton is not labeled in S15a in the spectrum of the strained WSe₂ with PMMA (blue). Also, excitation power dependence was not studied, and the assignment of the exciton, trion, excitons, and defects are purely based on their energies. The authors might need to be careful here as these energies can be affected by the dielectric environment and strain, varying from sample to sample. Therefore, direct comparison to references might not always be correct. Finally, the PL from Fig. S15a clearly shows that the PL is dominated by the defects, as the authors also mentioned.

Our Response:

We sincerely appreciate the reviewer's valuable comments and would like to acknowledge the concerns regarding the origin of the enhancement of the PL dominated by the defects affected by strain. It is true that structural changes due to both defects and strain significantly impact the optical properties of 2D vdW materials, necessitating a deep understanding to accurately probe excitonic species. While we acknowledge that the formation energy of defect structures can either increase or decrease with strain configuration, we would like to emphasize the prominence of the A exciton resonances at room temperature which dominates over defect state peaks. Additionally, the differential reflectance data, which exclusively reflects direct transitions rather than defect-related contributions, further substantiates this point, as we will explain henceforth.

First, we would like to address the potential contributions of other exciton species in comparison to neutral excitons at room temperature. In response to reviewer's suggestion, we have thoroughly re-examined the low-temperature PL spectrum and have now clearly labeled the neutral exciton peak (X^0) and trion peak (T) in the spectrum of strained WSe₂ with PMMA (blue). Additionally, we successfully confirmed these peak assignments by comparing samples with the same dielectric environment (PMMA/WSe₂/PDMS) at two different temperatures (RT and low-temperature).

At cryogenic temperatures (Figure S16a), a weak peak is observed from strained WSe₂ at an energy of 75 meV lower than the relatively distinct X^0 peak of unstrained WSe₂, suggesting that this is the weakened neutral exciton of strained WSe₂. It is noteworthy that at room temperature (Figure S16b), the X^0 peak in strained WSe₂ shows a similar red-shift of 75 meV compared to unstrained WSe₂. In addition, we can see the X^0 peak blue-shifted by 78.5 meV at 15 K

compared to the X^0 peak at 298 K. As the temperature decreases, due to the electron-phonon interaction decrease and lattice contraction effect, the energy band increases, resulting in the central wavelength of the neutral excitons to be blue-shifted in the range of 100 meV^[1].

There are several reasons why the neutral exciton peaks in strained WSe₂ at cryogenic temperatures remain relatively weak, as outlined below. Firstly, the significant decrease in neutral exciton (A^0) PL emission intensity in TMDs at cryogenic temperatures can primarily be attributed to the efficient trapping of excitons by defect states under lower thermal fluctuations, the activation of non-radiative relaxation channels, and comparatively enhanced exciton interactions with charge carriers (trions) and other excitons (e.g., formation of biexcitons or exciton annihilation)^[2,3]. Specifically, the reduced interaction of defect sites with phonons at low temperature leads to the radiative pathway of localized excitons being stronger than the non-radiative one, and even surpassing the intensity of neutral exciton.

Additionally, although neutral excitons in strained WSe₂ at cryogenic temperatures exhibit a red shift compared to the unstrained sample, the tensile strain on WSe₂ does not significantly enhance the PL intensity of the neutral exciton, unlike the observation at room temperature. This discrepancy may be attributed to the dominant conversion of neutral excitons to other states, particularly at cryogenic temperatures, and to alterations in exciton-phonon interactions under strain. This observation suggests that while defect-bound states may be present even at room temperature, the dominance of non-radiative pathways results in the neutral exciton being the predominant feature in the PL intensity.

The weakened neutral exciton peaks at cryogenic temperatures, attributed to enhanced radiative pathways from defect states, alongside the consistent neutral exciton peak shift (75 meV) compared to room temperature, support our observation of the dominant contribution of

neutral excitons at room temperature. This result supports the clear observation of neutral exciton funneling during valley pumping at room temperature, where the potential energy gradient causes excitons to drift to the apex and recombine radiatively with lower fluorescence energies.

Based on this discussion, we have revised Figure S16 with inclusion of the exciton and trion peaks, and added the explanation in manuscript as followed.

Revisions:

In Supplementary Information (Supplementary Text 2),

At cryogenic temperatures (Supplementary Fig. 16a), a weak peak is observed from strained WSe₂ at an energy of 75 meV lower than the relatively distinct X⁰ peak of unstrained WSe₂, suggesting that this is the weakened neutral exciton of strained WSe₂. It is noteworthy that at room temperature (Supplementary Fig. 16b), the X⁰ peak in strained WSe₂ shows a similar red-shift of 75 meV compared to unstrained WSe₂. In addition, we can see the X⁰ peak blue-shifted by 78.5 meV at 15 K compared to the X⁰ peak at 298 K. As the temperature decreases, lattice contraction occurs and the electron-phonon interaction decreases. Consequently, the energy band gap increases, resulting in the central wavelength of the neutral excitons to be blue-shifted in the range of 100 meV¹⁶. The neutral exciton peaks weaken at cryogenic temperatures due to enhanced radiative pathways from defect states. This, alongside the consistent neutral exciton peak shift (75 meV) compared to room temperature, supports our observation of the dominant presence and contribution of neutral excitons at room temperature.

Fig. S16. PL emission of unstrained/strained WSe₂ with and without PMMA thin film at varying temperature. a, PL emission spectra measured at 15 K. **b,** PL emission spectra measured at 298 K. Neutral exciton (X^0) and trion (T) were indexed in the plot. For the unstrained PL emission, we used the same monolayer WSe₂ flake with and without PMMA coating at both temperatures, while we employed different strained WSe₂ flakes due to wrinkle formation.

Comment 2: In response to my comment 3, the authors claimed that at room temperature, PL from biexcitons and defects are not relevant and PL at room temperature is mainly from bright exciton. I would disagree with that. The thermal fluctuation at room temperature is only ~ 26 meV, lower than the binding energy of some excitons such as biexciton. From the energy of view, some of the excitons might still be relevant. But more importantly, it is not clear why the PL from the defects can be quickly dismissed. The PL is significantly broadened at room temperature, and eventually, the defect PL cannot be distinguished from the bright exciton in the PL spectra. I suspect that defect PL contributed significantly to the room temperature PL the author observed, which seems to agree with the long lifetime the authors measured in the main text.

Our Response:

We sincerely appreciate the referee's valuable insights on validating the strain-induced exciton funneling phenomena, which could be complicated by the presence of biexcitons and defect effects. We fully agree that a careful characterization of the PL results is essential to differentiate the effects of defects affected by the strain from the exciton funneling from the local band structure modulation by strain. Given that both strain and defects can significantly alter the electronic and optical band structures, it is crucial to understand exciton funneling while considering potential influence from defects. However, we believe that the neutral exciton funneling observed at room temperature is in fact primarily supported by additional data, differential reflectance measurements, which provide direct evidence of neutral exciton transitions. As such, we will highlight how we used the differential reflectance spectrum to determine strain to be the primary factor influencing the room temperature exciton resonance.

The optical absorption spectrum derived from the differential reflectance resonates with direct transitions, providing more comprehensive information on charge transitions, which is advantageous for measuring the energies absorbed by electrons during interband transitions^[8–10]. In contrast, PL primarily reflects indirect transitions and is more sensitive to radiative recombination during the transition to the lowest energy state, thereby revealing localized defects and other excitonic complexes along with neutral excitons. To explore this further, we compared the differential reflectance spectra and PL spectra collected using the same confocal setup at room temperature, as shown in newly added Figure S20.

As demonstrated in Figure S20a-c, the flat WSe₂ exhibits a PL energy peak at around 1.66 eV, corresponding to the A exciton, which matches the differential reflectance peak. Additionally, the wrinkled WSe₂ shows red-shifted peak at the apex and blue-shifted peak at the valley in both PL and reflectance spectra. The coincidence in peak positions observed in differential reflectance and PL further supports the idea that **room temperature PL is substantially dominated by neutral excitons originating from the direct transitions.**

Furthermore, the spatially-resolved hyperspectral linescan images of PL and 2nd derivative of differential reflectance (Figure S20d and e) reveal distinct changes in the PL and reflectance peaks along the distance from the apex. The similar modulation of exciton energy at the apex in both reflectance and PL spectra reinforces our conclusion that primary PL peaks observed at room temperature originates from the direct transitions, supported by neutral exciton absorption (Figure S20f).

In addition, similar trends of PL shifts at apex and valley in other types of TMDs (e.g., MoSe₂, WS₂) on the wrinkle structure (newly added Figure S7) supports the structural and mechanical tunability of neutral exciton, allowing the modulation of local band structure. While

intrinsic defects may naturally form in WSe₂ and could be deconvoluted in the PL peak at room temperature, we strongly believe that these defects do not significantly affect our observations of neutral excitons funneling in room temperature via strain-induced band gap modulation.

Based on this discussion, we have revised our manuscript as the following:

Revisions:

In Manuscript,

While intrinsic defects may naturally form in WSe₂ and could be deconvoluted in the PL peak at room temperature, these defects do not significantly affect our observations of neutral excitons funneling in room temperature via strain-induced band gap modulation. The dominance of A exciton resonances at room temperature, surpassing defect state peaks, has been confirmed through a comparison of room-temperature PL data with both cryogenic PL and room-temperature differential reflectance spectroscopy (Supplementary Text 2).

...Wrinkling-induced modulation of local band structure was also demonstrated in other types of TMDs (e.g., MoSe₂, WS₂), showing similar trends of PL shifts at apex and valley of the wrinkle structure. (Supplementary Fig. 7). These results indicate that our soft wrinkle architecture enables structural tunability and mechanical reconfigurability of neutral excitons by controlling the local strain.

Figure S7. Wrinkle structure prepared with different transition metal dichalcogenides monolayers. a, Optical microscope image of wrinkled monolayer MoSe₂. b, Hyperspectral imaging in the red squared regime in a. (top) PL peak wavelength map, (middle) PL intensity map for 775 nm wavelength emission. (bottom) PL intensity map for 810 nm emission. c, PL emission spectra of wrinkled MoSe₂. d, Optical microscope image of wrinkled monolayer WS₂. e, PL peak wavelength map obtained by hyperspectral PL imaging. f, PL emission spectra of wrinkled WS₂.

In Supplementary Information (Supplementary Text 2),

To investigate governing role of neutral excitons at room temperature in the wrinkled WS₂, we carried out differential reflectance spectroscopy (Supplementary Fig. 20). In general, the optical absorption spectrum derived from the differential reflectance resonates with direct transitions, providing more comprehensive information on charge transitions, which is advantageous for measuring the energies absorbed by electrons during interband transitions^{17–19}.

In contrast, PL primarily reflects indirect transitions and is more sensitive to radiative recombination during the transition to the lowest energy state, thereby revealing localized defects and other excitonic complexes along with neutral excitons.

To this end, white light illumination from a halogen lamp was directed to the wrinkle sample through an iris aperture and then the reflected light was collected by confocal setup equipped with galvo-mirror scanning system. The differential reflectance spectra were recorded from unstrained flat WSe₂ and wrinkled WSe₂ at apex and valley (Supplementary Fig. 20a-c). For the flat WSe₂, PL and reflectance presented similar energy for A exciton at around 1.66 eV whereas reflectance revealed the B exciton at 2.1 eV due to spin-orbit splitting at the valence band.²⁰ In the case of wrinkled WSe₂, reflectance spectra were red- and blue-shifted at the apex and valley, respectively, compared with that of unstrained, flat WSe₂ (similar to the observation in PL spectroscopy), indicating that PL and reflection arose from the same excitonic transition.

Furthermore, the spatially-resolved hyperspectral linescan images of PL and 2nd derivative of differential reflectance (Supplementary Fig. 20d-f) reveal distinct changes in the PL and reflectance peaks along the distance from the apex. It should be noted that the energy difference between PL and reflectance observed for our wrinkled WSe₂ is less than 6 meV, which is distinct from the energy shift of other excitonic components such as charged trions (30 meV),²¹ biexcitons (50-60 meV),²² or defect-localized excitons (>60 meV).^{23,24}

While intrinsic defects may naturally form in WSe₂ and could be deconvoluted in the PL peak at room temperature, we strongly believe that these defects do not significantly affect our observations of neutral excitons funneling in room temperature via strain-induced band gap modulation. As it has been demonstrated, the alignment in peak position observed in differential

reflectance and PL further supports the notion that room temperature PL in this work is substantially dominated by neutral A excitons originating from the direct transitions.

Figure S20. PL and differential reflectance ($\Delta R/R$) spectra obtained from (a) flat WSe_2 , (b) wrinkled WSe_2 at apex, and (c) wrinkled WSe_2 at valley. **d-e**, Hyperspectral line scan of (a) PL and (b) second derivative of differential reflectance across wrinkle structure. **f**, Modulation of exciton energy measured by hyperspectral imaging.

Comment 3: In response to my previous comment 4, Fig. S17 actually shows significantly enhanced defect PL in the strained WSe₂. This again confirms my suspicion that the enhanced PL is from strain modulation of the defects.

Comment 4: In response to my previous comment 5, the authors agree that their sample quality is not superior and has weak low-temperature PL. I disagree with the argument that sample quality is not relevant here. If the enhanced PL is from the strain modulation of the defects (PL associated with at), it will change the interpretation of exciton funneling.

In summary, based on the quality of the current data, I suspect that the most enhanced PL is from defects instead of exciton funneling. In fact, I went through the manuscript carefully again and found that most of the data can be explained by the enhanced defect PL by strain. The measurement of PL away from the excitation spot could provide helpful information, but the data also need to be analyzed carefully, considering the diffraction limit, complicated topography of the structure, and the low quality of the WSe₂ (low mobility and low diffusion constant). For example, what will be the diffusion constant associated with the strain modulated WSe₂ and the plain WSe₂? To confirm the exciton funneling interpretation, the authors should consider performing their experiments at low temperatures, where they can separate the exciton from defects and other contributions.

Our Response:

We sincerely appreciate the reviewer's insightful comments regarding the enhanced PL in the apex regions. As the reviewer correctly pointed out, we acknowledge the dominant role of defect states and the potential enhancement of PL due to strain modulation of these defects at cryogenic

temperatures. However, as we noted in our previous responses, the similarity between the peak positions in the differential reflectance and PL spectra suggests that the room temperature PL spectra are strongly dominated by neutral excitons.

To further investigate the potential effects of defects on exciton funneling, we conducted pump-probe scanning at cryogenic temperatures (new Figure S19). This experiment at cryogenic temperature revealed less pronounced exciton funneling compared to our room temperature results (Figure 2). Specifically, we observed a slightly less elongated confinement of the excitons at the apex region. Additionally, we detected a very subtle exciton drift from the valley region towards the apex.

The observation of exciton funneling, even at low temperatures where radiative recombination might be dominated by localized defect sites, provides compelling evidence for strain gradient-induced bandgap modulation, which enhances PL and shifts the PL energy peaks. However, the weaker features of funneling at low temperatures can be interpreted in two ways: (1) the enhanced radiative recombination channels of localized defect sites at cryogenic temperatures may inhibit the drift of neutral excitons, or (2) the dominant localized excitons at these temperatures may not drift as efficiently as neutral excitons.

Furthermore, the larger diffusion coefficient in strained WSe₂, influenced by the strain field compared to the unstrained material, can be inferred from the simulated flux of steady-state PL emission (Figure 3h). This finding aligns with our experimental PL emission observed in Figures 2d and 3g and clearly indicates that the strain-induced energy gradient plays a significant role in influencing parameters critical to exciton transport, including diffusion, drift, and mobility.

To reflect the referee's comments, we have added further results revealing the exciton funneling at cryogenic temperature, and added explanations in the supporting information for clarification. Specifically, we have emphasized that the optical tuning of fluorescence at room temperature is primarily governed by the coupling between lattice strain and charge-neutral excitons, rather than by varied exciton components or out-of-plane geometry.

Revisions:

In Supplementary Information (Supplementary Text 2),

~~The enhanced PL intensity and redshifted PL peak at the apex, compared to the valley, regardless of the focal plane (Figure S2), indicate that the relative stability of bright and dark excitons varies depending on whether the strain is tensile or compressive.~~ **The observation of exciton funneling, even at low temperatures (Supplementary Fig. 19), provides compelling evidence of strain gradient-induced bandgap modulation, which enhances PL and shifts the PL energy peaks. The less prominent features of funneling at low temperatures are attributed to the fact that either (1) the enhanced radiative recombination channels of localized defect sites at cryogenic temperatures may inhibit the drift of neutral excitons, or (2) the dominant localized excitons at these temperatures may not drift as efficiently as neutral excitons. These results suggest that PL enhancement with a red shift of the emission peak at the apex in locally strained WSe₂ at room temperature are unlikely dominated by other excitonic components. Instead, these effects may be driven by strain gradient-induced changes in the neutral A exciton, such as an energy gradient in the excitation area or exciton funneling.**

Figure S19. Demonstration of exciton funneling at cryogenic temperature. Pump-probe scanning maps with excitation laser fixed at left apex, valley, and right apex, respectively.

RESPONSE TO REVIEWERS' COMMENTS

Reviewer #2 (Remarks to the Author):

I have read the authors' reply thoroughly, and I would say I appreciate the authors' persistence and the extra efforts in getting additional data on the reflectance spectra as well as the new data on possible low-temperature exciton funneling. Although I still have some reservations about the interpretation of defect contributions, I believe that the current manuscript contains enough information for the community, and I would recommend the publication in Nature Communications.

I still have a few comments for the authors to consider, but this does not change my recommendation and does need to be addressed in this current version.

Our Response: We are grateful to the reviewer for his/her positive evaluation of our manuscript and publication recommendation in Nature Communications.

1) The defect contribution at room temperature is complicated due to the broad PL width. The argument that defects do not play a major role at room temperature might not be exactly correct. The PL width is broader than the thermal fluctuations, and the defect could have communication with the exciton state.

Our Response: We appreciate the reviewer's comment regarding the broad PL width and defect. While we agree that the defect contribution to PL at room temperature is present and might be worthy of additional investigation, we believe the strain-gradient induced drift of neutral exciton (i.e., exciton funneling) at room temperature is in fact the dominant mechanism supported by the coincidence in peak positions between differential reflectance and PL. The consistent redshift of the neutral exciton peak observed at the strained WSe₂ apex even at cryogenic temperatures,

provides strong evidence for the predominant presence and significant contribution of neutral excitons at room temperature.

2) The reflection spectra are a strong point to argue the contribution of the exciton funneling.

However, if the authors look at the spectra closely, the resonance of reflection does not exactly align with the PL. Analysis of the difference could be useful.

Our Response: We are thankful for the reviewer's comment. In our revised Supplementary Information, we have provided additional discussions regarding the analysis of PL and reflectance.

Revisions: In Supplementary Information (Supplementary Text 2)

The energy difference observed between reflectance and emission is typically attributed to the Stokes shift, where the emitted wavelength is longer than the absorbed wavelength due to energy loss. For WSe₂, the Stokes shift is reported to be around 3 meV²⁵, while our WSe₂ wrinkled structure shows a slightly larger but still minimal shift of 6 meV, suggesting minimal defect influence on the strained WSe₂.

3) The description of the low-temperature exciton funneling is lacking details. I sort of guessed what the authors wanted to convey, but it was not clear directly from the description.

Our Response: We are thankful for the reviewer's comment. In our revised Supplementary Information, we have provided additional discussions regarding the low-temperature exciton funneling.

Revisions: In Supplementary Information (Supplementary Text 2)

The observation of exciton funneling, even at low temperatures (Supplementary Fig. 19), displays a similar pattern to the pump-probe PL maps at room temperature as a function of the excitation position (Supplementary Fig. 9). Specifically, when the excitation laser is positioned at the apex, emission predominantly arises from the apex. In contrast, when the excitation laser is set at the valley, emission is still probed from the apex region due to exciton funneling toward the apex.